# Benchmarks and Algorithms for Offline Preference-Based Reward Learning

**Daniel Shin**                                                    *danielshin@cs.stanford.edu*
*Computer Science Department*
*Stanford University*

**Anca D. Dragan**                                                 *anca@berkeley.edu*
*EECS Department*
*University of California, Berkeley*

**Daniel S. Brown**                                                *dsbrown@cs.utah.edu*
*School of Computing*
*University of Utah*

**Reviewed on OpenReview:** *https://openreview.net/forum?id=TGuXXlbKsn*

## Abstract

Learning a reward function from human preferences is challenging as it typically requires having a high-fidelity simulator or using expensive and potentially unsafe actual physical rollouts in the environment. However, in many tasks the agent might have access to offline data from related tasks in the same target environment. While offline data is increasingly being used to aid policy optimization via offline RL, our observation is that it can be a surprisingly rich source of information for preference learning as well. We propose an approach that uses an offline dataset to craft preference queries via pool-based active learning, learns a distribution over reward functions, and optimizes a corresponding policy via offline RL. Crucially, our proposed approach does not require actual physical rollouts or an accurate simulator for either the reward learning or policy optimization steps. To test our approach, we first evaluate existing offline RL benchmarks for their suitability for offline reward learning. Surprisingly, for many offline RL domains, we find that simply using a trivial reward function results in good policy performance, making these domains ill-suited for evaluating learned rewards (even those learned by non preference-based methods). To address this, we identify a subset of existing offline RL benchmarks that are well suited for reward learning and also propose new offline reward learning benchmarks which allow for more open-ended behaviors allowing us to better test offline reward learning from preferences. When evaluated on this curated set of domains, our empirical results suggest that combining offline RL with learned human preferences can enable an agent to learn to perform novel tasks that were not explicitly shown in the offline data.

## 1 Introduction

For automated sequential decision making systems to effectively interact with humans in the real world, these systems need to be able to safely and efficiently adapt to and learn from different users. Apprenticeship learning (Abbeel & Ng, 2004)—also called learning from demonstrations (Argall et al., 2009) or imitation learning (Osa et al., 2018)—seeks to allow robots and other autonomous systems to learn how to perform a task through human feedback. Even though traditional apprenticeship learning uses expert demonstrations (Argall et al., 2009; Osa et al., 2018; Arora & Doshi, 2021), preference queries—where a user is asked to compare two trajectory segments—have been shown to more accurately identify the reward (Jeon et al., 2020) while reducing the burden on the user to generate near-optimal actions (Wirth et al., 2017; Christiano et al.,

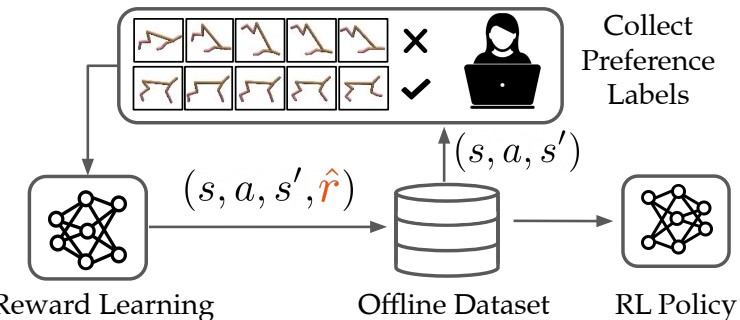

Figure 1: Offline Preference-Based Reward Learning (OPRL) enables safe and efficient preference-based policy customization without any environmental interactions. Given an offline database consisting of trajectories, OPRL queries for pairwise preference labels over trajectory segments from the database, learns a reward function from these preference labels, and then performs offline RL using the learned reward function.

2017). However, one challenge is that asking these queries requires either executing them in the physical world (which can be unsafe), or executing them in a simulator and showing them to the human (which requires simulating the world with high enough fidelity).

Learning a policy via RL has the same challenges of needing a simulator or physical rollouts (Sutton & Barto, 2018). To deal with these problems, offline RL has emerged as a promising way to do policy optimization with access to *offline* data only (Levine et al., 2020). Our key observation in this paper is that an analogous approach can be applied to reward learning: rather than synthesizing and executing preference queries in a simulator or in the physical world, we draw on an offline dataset to extract segments that would be informative for the user to compare. Since these segments have already been executed and recorded by the agent, all we need to do is replay them to the human, and ask for a comparison.

We call this approach Offline Preference-based Reward Learning (OPRL), a novel approach that consists of offline preference-based reward learning combined with offline RL. Our approach is summarized in Figure 1. OPRL has two appealing and desirable properties: safety and efficiency. No interactions with the environment are needed for either reward learning or policy learning, removing the sample complexity and safety concerns that come from trial and error learning in the environment.

Although the individual components of our framework are not novel by themselves, our novel contribution lies in the fundamentally new and unexplored capability of learning unseen tasks during test time. For example as seen in Figure 2, even if all demonstrations are short segments of random point to point navigation, we demonstrate that OPRL can recover a policy that is able to go in a counter-clockwise orbit around the entire maze infinitely. The key to achieving this is the ability to stitch together incomplete segments from the original dataset to create one long trajectory for a new task during test time.

By actively querying for comparisons between *segments* (or snippets) from a variety of different rollouts, the agent can gain valuable information about how to customize it's behavior based on a user's preferences. This is true even when none of the data in the offline dataset explicitly demonstrates the desired behavior. For example, in Figure 2, an agent learns an orbiting behavior from human preferences, despite never having seen a complete orbit in the offline data.

To effectively study offline reward learning, we require a set of benchmark domains to evaluate different approaches. However, while there are many standard RL benchmarks, there are fewer benchmarks for reward learning or even imitation learning more broadly. Recent work has shown that simply using standard RL benchmarks and masking the rewards is not sufficiently challenging since often learning a +1 or -1 reward everywhere is sufficient for imitating RL policies (Freire et al., 2020). While there has been progress in developing imitation learning benchmarks (Toyer et al., 2020; Freire et al., 2020), existing domains assume an online reward learning and policy optimization setting, with full access to the environment, which is only realistic in simulation due to efficiency and safety concerns.

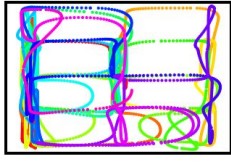
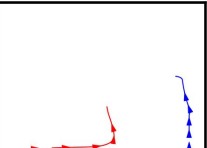
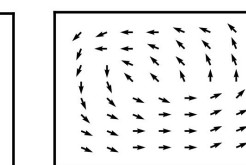
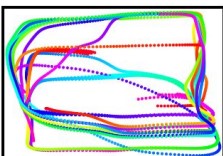

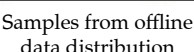

| Samples from offline data distribution | Examples of pairwise preference queries selected from offline data | States (position & velocity) with highest learned reward. | Trajectories from learned policy |

Figure 2: OPRL selects active queries from an offline dataset consisting of random point to point navigation. These active queries elicit preferences about a human's preferences (the human wants the robot to perform counter-clockwise orbits and prefers blue over red trajectories). Note that none of the segments in the offline dataset consists of a complete counter-clockwise orbit. However, OPRL is able to recover a counter-clockwise orbiting policy given human preferences. The learned reward function matches the human's preferences and, when combined with offline RL, leads to an appropriate policy that looks significantly different from the original offline data distribution. Because offline data is often not collected for the specific task a human wants, but for other tasks, being able to repurpose data from a variety of sources is important for generalizing to different user preferences in offline settings where we cannot easily just gather lots of new online data.

One of our contributions is to evaluate a variety of existing offline RL benchmarks (Fu et al., 2020) in the offline reward learning setting, where we remove access to the true reward function. Surprisingly, we find that many offline RL benchmarks are ill-suited for studying and comparing different reward learning approaches—simply replacing all actual rewards in the offline dataset with zeros, or a constant, results in performance similar to, or better than, the performance obtained using the true rewards! This is problematic since it means that high performance in these domains is not always indicative of a better learned reward—rather it appears that performance in many domains in mainly affected by the quality of the data (expert vs suboptimal) and the actual reward value has little impact on offline RL. To better isolate and study the effect of different reward learning approaches, we identify a subset of environments where simply using a trivial reward function guess results in significant degradation in policy performance, motivating the use of these environments when evaluating reward learning methods. We identify high offline data variability and multi-task data as two settings where reward learning is necessary for good policy learning.

Because most existing offline RL domains are not suitable for studying the effects of reward learning, we also propose several new tasks designed for open-ended reward learning in offline settings. Figure 2 shows an example from our experiments where the learning agent has access to a dataset composed of trajectories generated offline by a force-controlled pointmass robot navigating between random start and goal locations chosen along a 5x3 discrete grid. OPRL uses this dataset to learn to perform counter-clockwise orbits, a behavior never explicitly seen in the offline data. Figure 2 shows a sequence of active queries over trajectory snippets taken from the offline dataset. The blue trajectories were labeled by the human as preferable to the red trajectories. We also visualize the learned reward function. Using a small number of queries and a dataset generated by a very different process than the desired task, we are able to learn the novel orbiting behavior in a completely offline manner.

We summarize the contributions of this paper as follows:

1. We propose and formalize the novel problem of offline preference-based reward learning.

2. We evaluate a large set of existing offline RL benchmarks, identify a subset that is well-suited for evaluating offline reward learning, and propose new benchmarks that allow for more open-ended behavior.

3. We perform an empirical evaluation of OPRL, comparing different methods for maintaining uncertainty and comparing different acquisition functions that use the uncertainty estimates to actively select informative queries. We provide evidence that ensemble-based disagreement queries outperform other approaches.

4. Our results suggest that there is surprising value in offline datasets. The combination of offline reward learning and offline RL leads to highly efficient reward inference and enables agents to learn to perform tasks not explicitly demonstrated in the offline dataset.

## 2 Related Work

Prior work on preference-based reward learning typically focuses on online methods that require actual rollouts in the environment or access to an accurate simulator or model (Wirth et al., 2017; Christiano et al., 2017; Sadigh et al., 2017; Biyik & Sadigh, 2018; Brown et al., 2019; 2020b; Lee et al., 2021). However, accurate simulations and model-based planning are often not possible. Furthermore, even when an accurate simulator or model is available, there is the added burden of solving a difficult sim-to-real transfer problem (Tobin et al., 2017; Peng et al., 2018; Chebotar et al., 2019). In most real-world scenarios, the standard preference learning approach involving many episodes of trial and error in the environment is likely to be unacceptable due to safety and efficiency concerns.

Prior work on safe apprenticeship learning either enables learners to estimate risky actions (Zhang & Cho, 2016) and request human assistance (Hoque et al., 2021), optimizes policies for tail risk rather than expected return (Lacotte et al., 2019; Javed et al., 2021), or provides high-confidence bounds on the performance of the agent's policy when learning from demonstrations (Brown & Niekum, 2018; Brown et al., 2020a); however, these methods all rely on either an accurate dynamics model or direct interactions with the environment. By contrast, our approach towards safety is to develop a fully offline apprenticeship learning algorithm to avoid costly and potentially unsafe physical data collection during reward and policy learning.

While there has been some work on offline apprenticeship learning, prior work focuses on simple environments with discrete actions and hand-crafted reward features (Klein et al., 2011; Bica et al., 2021) and requires datasets that consist of expert demonstrations that are optimal for a particular task (Lee et al., 2019). Other work has considered higher-dimensional continuous tasks, but assumes access to expert demonstrations or requires experts to label trajectories with explicit reward values (Cabi et al., 2019; Zolna et al., 2020). By contrast, we focus on fully offline reward learning via small numbers of qualitative preference queries which are known to be much easier to provide than fine-grained reward labels or near-optimal demonstrations (Kendall, 1948; Stewart et al., 2005; Saaty, 2008; Wirth et al., 2017). Prior work by Castro et al. (2019) is similar to ours in that they learn a reward function from offline ranked demonstrations of varying qualities. However, their proposed approach is only evaluated on MDPs with finite state spaces where they learn a unique reward for each state by solving a quadratic program. By contrast, our approach learns reward features from low-level, continuous state information and uses a differentiable pairwise preference loss function rather than quadratic programming.

Other prior work has considered offline imitation learning. Methods such as behavioral cloning are trained on offline expert demonstrations, but suffer from compounding errors (Pomerleau, 1988). DemoDICE (Kim et al., 2021) seeks to imitate expert demonstrations that are provided offline and improves stability by also leveraging a dataset of suboptimal demonstrations. By contrast, our approach does not require expert demonstrations, just an offline dataset of state transitions and learns an explicit reward function from preferences. IQ-Learn (Garg et al., 2021) is capable of both offline and online imitation learning. Rather than learning a reward function, they learn a parameterized Q-function. Similar to DemoDICE, IQ-Learn requires access to expert demonstrations, whereas we only require human preference labels over offline data of any quality—some of our experiments use data generated from a completely random policy. The main downside to imitation learning methods like DemoDICE and IQ-Learn is the requirement of having expert demonstrations. We avoid this problem by learning rewards from preference labels over pairs of trajectory segments, something that is easily provided, even for humans who cannot provide expert demonstrations.

There has also been previous work that focused on apprenticeship learning from heterogeneous human demonstrations in resource scheduling and coordination problems (Paleja et al., 2019). Our approach also works by learning from heterogeneous data, but uses preference queries to learn a reward function which is then used for offline RL algorithms rather than learning a scheduling algorithm from human demonstrations.

## 3    Problem Definition

We model our problem as a Markov decision process (MDP), defined by the tuple $(S, A, r, P, \rho_0, \gamma)$, where $S$ denotes the state space, $A$ denotes the action space, $r : S \times A \to \mathbb{R}$ denotes the reward, $P(s'|s, a)$ denotes the transition dynamics, $\rho_0(s)$ denotes the initial state distribution, and $\gamma \in (0, 1)$ denotes the discount factor.

In contrast to standard RL, we do not assume access to the reward function $r$. Furthermore, we do not assume access to the MDP during training. Instead, we are provided a static dataset, $\mathcal{D}$, consisting of trajectories, $\mathcal{D} = \{\tau_0, ..., \tau_N\}$, where each trajectory $\tau_i$ consists of a contiguous sequence of state, action, next-state transitions tuples $\tau_i = \{(s_{i,0}, a_{i,0}, s'_{i,0}), ..., (s_{i,M}, a_{i,M}, s'_{i,M})\}$. Unlike imitation learning, we do not assume that this dataset comes from a single expert attempting to optimize a specific reward function $r(s, a)$. Instead, the dataset $\mathcal{D}$ may contain data collected randomly, data collected from a variety of policies, or even from a variety of demonstrations for a variety of tasks.

Instead of having access to the reward function, $r$, we assume access to an expert that can provide a small number of pairwise preferences over trajectory snippets from an offline dataset $\mathcal{D}$. Given these preferences, the goal is to find a policy $\pi(a|s)$ that maximizes the expected cumulative discounted rewards (also known as the discounted returns), $J(\pi) = \mathbb{E}_{\pi, P, \rho_0} \left[ \sum_{t=0}^{\infty} \gamma^t r(s_t, a_t) \right]$, under the unknown true reward function $r$ in a fully offline fashion—we assume no access to the MDP other than the offline trajectories contained in $\mathcal{D}$.

## 4    Offline Preference-Based Reward Learning

We now discuss our proposed algorithm, Offline Preference-based Reward Learning (OPRL). OPRL is an offline, active preference-based learning approach which first searches over an offline dataset of trajectories and sequentially selects new queries to be labeled by the human expert in order to learn a reward function that models the user's preferences and can be used to generate a customized policy based on a user's preferences via offline RL.

As established by a large body of prior work (Christiano et al., 2017; Ibarz et al., 2018; Brown et al., 2019; Palan et al., 2019), the Bradley-Terry pairwise preference model (Bradley & Terry, 1952) is an appropriate model for learning reward functions from user preferences over trajectories. Given a preference over trajectories, $\tau_i \prec \tau_j$, we seek to maximize the probability of the preference label:

$$P(\tau_i \prec \tau_j \mid \theta) = \frac{\exp \sum_{s \in \tau_j} \hat{r}_\theta(s)}{\exp \sum_{s \in \tau_i} \hat{r}_\theta(s) + \exp \sum_{s \in \tau_j} \hat{r}_\theta(s)}, \tag{1}$$

by approximating the reward at state $s$ using a neural network, $\hat{r}_\theta(s)$, such that $\sum_{s \in \tau_i} \hat{r}_\theta(s) < \sum_{s \in \tau_j} \hat{r}_\theta(s)$ when $\tau_i \prec \tau_j$.

OPRL actively queries to obtain informative preference labels over trajectory snippets sampled from the offline dataset. In order to perform active learning, OPRL needs a way to estimate the uncertainty over the predicted preference label for unlabeled trajectory pairs, in order to determine which queries would be most informative (result in the largest reduction in uncertainty over the true reward function). In this paper, we investigate two different methods to represent uncertainty (ensembles and Bayesian dropout) along with two different acquisition functions (disagreement and information gain). We describe these approaches below.

### 4.1    Representing Reward Uncertainty

We compare two of the most popular methods for obtaining uncertainty estimates when using deep learning: ensembles (Lakshminarayanan et al., 2016) and Bayesian dropout (Gal & Ghahramani, 2016). On line 4 of Algorithm 1, we initialize an ensemble of reward models for ensemble queries or a single reward model for Bayesian dropout.

**Ensemble Queries**    Following work on online active preference learning work by Christiano et al. (2017), we test the effectiveness of training an ensemble of reward models to approximate the reward function posterior

using the Bradley-Terry preference model. Similar to prior work (Christiano et al., 2017; Reddy et al., 2020), we found that initializing the ensemble networks with different seeds was sufficient to produce a diverse set of reward functions.

**Bayesian Dropout** We also test the effectiveness of using dropout to approximate the reward function posterior. As proposed by Gal & Ghahramani (2016), we train a reward network using pairwise preferences and apply dropout to the last layer of the network during training. To predict a return distribution over candidate pairs of trajectories, we still apply dropout and pass each trajectory through the network multiple times to obtain a distribution over the returns.

## 4.2 Active Learning Query Selection

In contrast to prior work on active preference learning, we do not require on-policy rollouts in the environment (Christiano et al., 2017; Lee et al., 2021) nor require synthesizing queries using a model (Sadigh et al., 2017). Instead, we generate candidate queries by searching over randomly chosen pairs of sub-trajectories obtained from the offline dataset. Given a distribution over likely returns for each trajectory snippet in a candidate preference query, we estimate the value of obtaining a label for this candidate query. We consider two methods for computing the value of a query: disagreement and information gain. We then take the trajectory pair with the highest predicted value and ask for a pairwise preference label. On line 6 of Algorithm 1, we can either use disagreement or information gain as the estimated value of information (VOI).

**Disagreement** When using disagreement to select active queries, we select pairs with the highest ensemble disagreement among the different return estimates obtained from either ensembling or Bayesian dropout. Following Christiano et al. (2017), we calculate disagreement as the variance in the binary comparison predictions: if fraction $p$ of the posterior samples predict $\tau_i \succ \tau_j$ while the other $1 - p$ ensemble models predict $\tau_i \preceq \tau_j$, then the variance of the query pair $(\tau_i, \tau_j)$ is $p(1 - p)$.

**Information Gain Queries** As an alternative to disagreement, we also consider the expected information gain (Cover, 1999) between the reward function parameters $\theta$ and the outcome $Y$ of querying the human for a preference. We model our uncertainty using an approximate posterior $p(\theta \mid \mathcal{D})$, given by training an ensemble or dropout network on our offline dataset $\mathcal{D}$. Houlsby et al. (2011) show that the information gain of a potential query can be formulated as:

$$I(\theta; Y \mid \mathcal{D}) = H(Y \mid \mathcal{D}) - \mathbb{E}_{\theta \sim p(\theta \mid \mathcal{D})}[H(Y \mid \theta, \mathcal{D})]. \tag{2}$$

Intuitively, the information gain will be maximized when the first term is high, meaning that the overall model has high entropy, but the second term is low, meaning that each individual hypothesis $\theta$ from the posterior assigns low entropy to the outcome $Y$. This will happen when the individual hypotheses strongly disagree with each other and there is no clear majority. We approximate both terms in the information gain equation with samples obtained via ensemble or dropout. See Appendix D for further details.

**Searching the Offline Dataset for Informative Queries** We note that this process of searching for informative queries can be performed quite efficiently since the information gain or ensemble disagreement for each candidate query can be computed in parallel. Furthermore, we can leverage GPU parallelization by feeding all of the states in one or more trajectories into the reward function network as a batch. Finally, we note that searching for the next active query can be formulated as an any-time algorithm—we can continue to sample random pairs of trajectories from the offline dataset to evaluate and when a particular time or computation limit is reached and then simply return the most informative query found so far.

## 4.3 Policy Optimization

Given a learned reward function obtained via preference-learning, we can then use any existing offline or batch RL algorithm (Levine et al., 2020) to learn a policy without requiring knowledge of the transition dynamics or rollouts in the actual environment. The entire OPRL pipeline is summarized in Algorithm 1.

---

**Algorithm 1** OPRL

---

 1: **Require:** A dataset $\mathcal{D} = \{\tau_0, ..., \tau_N\}$, where each trajectory $\tau_i$ consists of $\tau_i = \{(s_{i,0}, a_{i,0}, s'_{i,0}), ..., (s_{i,M}, a_{i,M}, s'_{i,M})\}$.
 2: // REWARD LEARNING
 3: Generate dataset of pairs of snippets $\mathcal{D}_{snip}$
 4: Initialize reward model $\hat{r}_\psi$
 5: **for** each iteration **do**
 6:      Compute estimate of VOI across all pairs in $\mathcal{D}_{snip}$
 7:      Find snippet pair $(\tau_{s,1}, \tau_{s,2})$ with highest estimated VOI
 8:      Query preference label $y$
 9:      Store query pair and label $\mathcal{D}_{rew} \leftarrow (\tau_{s,1}, \tau_{s,2}, y)$
10:      Train reward model with updated $\mathcal{D}_{rew}$
11: **end for**
12: Label transitions in $\mathcal{D}$ with $\hat{r}_\psi$
13: // POLICY LEARNING
14: Run selected offline RL algorithm on $\mathcal{D}$

---

## 5 Experiments and Results

We first evaluate a variety of popular offline RL benchmarks from D4RL (Fu et al., 2020) to determine which domains are most suited for evaluating offline reward learning. Prior work on reward learning has shown that simply showing good imitation learning performance on an RL benchmark is not sufficient to demonstrate good reward learning (Freire et al., 2020). In particular, we seek domains where simply using an all zero or constant reward does not result in good performance. After isolating several domains where learning a shaped reward actually matters (Section 5.1), we evaluate OPRL on these selected benchmark domains and investigate the performance of different active query strategies (Section 5.2). Finally, we propose and evaluate several tasks specifically designed for offline reward learning (Section 5.3). To further analyze our method, we include sensitivity analyses on the number of initial queries, queries per round, ensemble models, and dropout samples (Appendix B, C)

Videos of learned behavior and code is available in the Supplement.

### 5.1 Evaluating Offline RL Benchmarks

An important assumption made in our problem statement is that we do not have access to the reward function. Before describing our approach for reward learning, we first investigate in what cases we actually need to learn a reward function. We study a set of popular benchmarks used for offline RL (Fu et al., 2020). To test the sensitivity of these methods to the reward function, we evaluate the performance of offline RL algorithms on these benchmarks when we set the reward equal to zero for each transition in the offline dataset and when we set all the rewards equal to a constant (the average reward over the entire dataset).

We evaluated four of the most commonly used offline RL algorithms: Advantage Weighted Regression (AWR) (Peng et al., 2019), Batch-Constrained deep Q-learning (BCQ) (Fujimoto et al., 2019), Bootstrapping Error Accumulation Reduction (BEAR) (Kumar et al., 2019), and Conservative Q-Learning (CQL) (Kumar et al., 2020). Table 1 shows the resulting performance across a large number of tasks from the D4RL benchmarks (Fu et al., 2020). We find that for tasks with only expert data, there is no need to learn a reward function since using a trivial reward function often leads to expert performance. We attribute this to the fact that many offline RL algorithms are similar to behavioral cloning (Levine et al., 2020). Thus, when an offline RL benchmarks consists of only expert data, offline RL algorithms do not need reward information to perform well. In Appendix E we show that AWR reduces to behavioral cloning in the case of an all zero reward function.

Conversely, when the offline dataset contains a wide variety of data that is either random or multi-task (e.g. the Maze2D environment consists of an agent traveling to randomly generated goal locations), we find

Table 1: Offline RL performance on D4RL benchmark tasks comparing the performance with the true reward to performance using a constant reward equal to the average reward over the dataset (Avg) and zero rewards everywhere (Zero). Even when all the rewards are set to the average or to zero, many offline RL algorithms still perform surprisingly well. In this table, we present the experiments ran with AWR. Results are averages over three random seeds. Bolded environments are ones where we find the degradation percentage to be over a chosen threshold and are used for later experiments with OPRL. The degradation percentage is calculated as $\frac{\max(GT-\max(AVG,ZERO,RANDOM),0)}{GT-\min(AVG,ZERO,RANDOM)} \times 100\%$ and the threshold is set to 20%. The degradation percentage is used to determine domains where trivial reward functions do not lead to good performance to isolate the effect of reward learning in later experiments.

| TASK | GT | AVG | ZERO | RANDOM | DEGRADATION % |
|---|---|---|---|---|---|
| FLOW-RING-RANDOM | -42.5 | -42.9 | -44.3 | -166.2 | 0.3 |
| **FLOW-MERGE-RANDOM** | 160.3 | 85.2 | 85.6 | 117.0 | 57.7 |
| **MAZE2D-UMAZE** | 104.4 | 56.5 | 55.0 | 49.5 | 87.3 |
| **MAZE2D-MEDIUM** | 134.6 | 30.5 | 34.5 | 44.8 | 86.3 |
| **HALFCHEETAH-RANDOM** | 11.0 | -49.4 | -48.3 | -285.8 | 20.0 |
| HALFCHEETAH-MEDIUM-REPLAY | 4138.2 | 3934.7 | 3830.2 | -285.8 | 4.6 |
| HALFCHEETAH-MEDIUM | 4096.0 | 3984.4 | 3898.3 | -285.8 | 2.5 |
| HALFCHEETAH-MEDIUM-EXPERT | 669.9 | 401.7 | 560.5 | -285.8 | 11.4 |
| HALFCHEETAH-EXPERT | 467.8 | 511.1 | 493.1 | -285.8 | 0.0 |
| HOPPER-RANDOM | 123.3 | 129.8 | 29.1 | 18.3 | 0.0 |
| HOPPER-MEDIUM-REPLAY | 1045.6 | 1127.4 | 954.6 | 18.3 | 0.0 |
| HOPPER-MEDIUM | 1152.1 | 1192.8 | 963.6 | 18.3 | 0.0 |
| HOPPER-MEDIUM-EXPERT | 623 | 615.8 | 587.3 | 18.3 | 1.2 |
| HOPPER-EXPERT | 571.7 | 617.2 | 427.6 | 18.3 | 0.0 |
| **KITCHEN-COMPLETE** | 0.3 | 0.2 | 0.3 | 0.0 | 25.0 |
| KITCHEN-MIXED | 0.3 | 0.5 | 0.3 | 0.0 | 16.7 |
| KITCHEN-PARTIAL | 0.3 | 0.2 | 0.3 | 0.0 | 0.0 |

that running offline RL algorithms on the dataset with an all zero or constant reward function significantly degrades performance.

## 5.2 Reward Learning on a Subset of D4RL

To evaluate the benefit of offline reward learning, we focus our attention on the D4RL benchmark domains shown in Figure 3 which showed significant degradation in Table 1 since these are domains where we can be confident that good performance is due to learning a good reward function. We define significant degradation to be degradation percentage over 20%, which was chosen to give enough room for OPRL to improve on in comparison to uninformative rewards like avg or zero masking. We selected both of the Maze environments, the Flow Merge traffic simulation, the Halfcheetah random environment, and the Franka Kitchen-Complete environment. To facilitate a better comparison across different methods for active learning, we use oracle preference labels, where one trajectory sequence is preferred over another if it has higher ground-truth return. We report the performance of OPRL using AWR (Peng et al., 2019), which we empirically found to work for policy optimization across the different tasks. In the appendix we also report results when using CQL (Kumar et al., 2020) for policy optimization.

### 5.2.1 Maze Navigation

We first consider the Maze2d domain (Fu et al., 2020), which involves moving a force-actuated ball (along the X and Y axis) to a fixed target location. The observation is 4 dimensional, which consists of the $(x, y)$ location and velocities. The offline dataset consists of one continuous trajectory of the agent navigation to

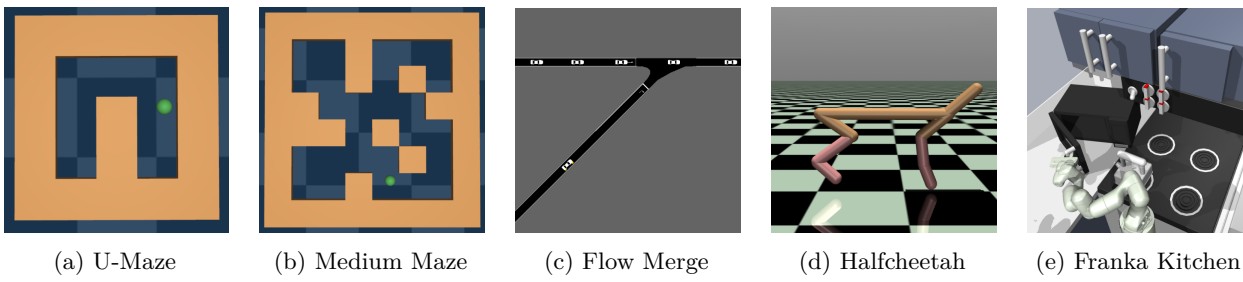

(a) U-Maze     (b) Medium Maze     (c) Flow Merge     (d) Halfcheetah     (e) Franka Kitchen

Figure 3: Experimental domains chosen from D4RL (Fu et al., 2020) for use in offline preference-based reward learning.

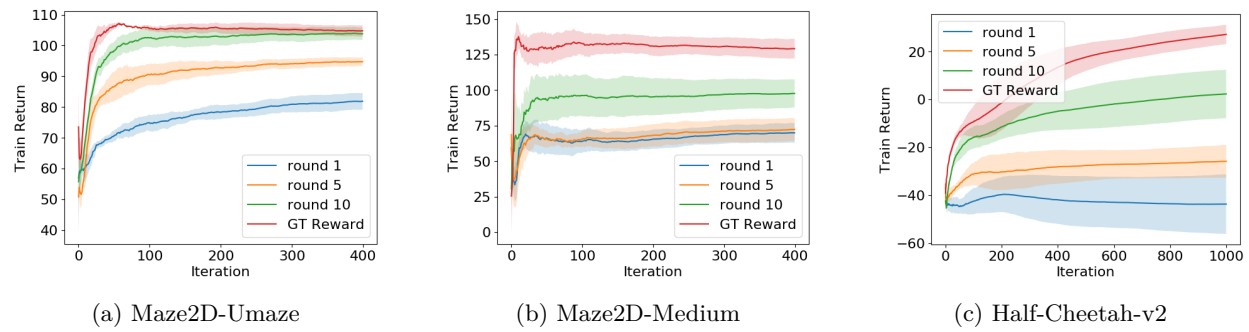

(a) Maze2D-Umaze     (b) Maze2D-Medium     (c) Half-Cheetah-v2

Figure 4: (a) Ensemble disagreement after 10 rounds of active queries (1 query per round) in Maze2d-Umaze. (b) Ensemble disagreement after 10 rounds of active queries (1 query per round) in Maze2d-Medium. (c) Ensemble disagreement after 10 rounds of active queries (10 queries per round) in Halfcheetah.

random intermediate goal locations. The true reward, which we only use for providing synthetic preference labels, is the negative exponentiated distance to a held-out goal location.

For our experimental setup, we first randomly select 5 pairs of trajectory snippets and train 5 epochs with our models. After this initial training process, for each round, one additional pair of trajectories is queried to be added to the training set and we train one more epoch on this augmented dataset. The learned reward model is then used to predict the reward labels for all the state transitions in the offline dataset, which is then used to train a policy via offline RL (e.g. AWR (Peng et al., 2019)). As a baseline, we also compare against a random, non-active query baseline.

The results are provided in Table 2. We found that ensemble disagreement is the best performing approach for both Maze2D-Umaze and Maze2D-Medium and outperforms T-REX. The offline RL performance for ensemble disagreement are shown in Figure 4. We find that ensemble disagreement on Maze2D-Umaze is able to quickly approach the performance when using the ground-truth reward function on the full dataset of 1 million state transitions after only 15 preference queries (5 initial + 10 active). Maze2D-Medium is more difficult and we find that 15 preference queries is unable to recover a policy on par with the policy trained with ground truth rewards. We hypothesize that this is due to the fact that Maze2D is goal oriented and learning a reward with trajectory comparisons might be difficult since the destination matters more than the path.

### 5.2.2 MuJoCo Tasks

We next tested OPRL on the MuJoCo environment Halfcheetah-v2. For the tasks, the agent is rewarded for making the Halfcheetah move forward as fast as possible. The MuJoCo tasks presents an additional challenge in comparison to the Maze2d domain since they have higher dimensional observations spaces, 17 for Halfcheetah-v2. The dataset contains 1 million transitions obtained by rolling out a random policy. Our experimental setup is similar to Maze2D, except we start with 50 pairs of trajectories instead of 5 and we

Table 2: **OPRL Performance Using Reward Predicted After N Rounds of Querying**: Offline RL performance using rewards predicted either using a static number of randomly selected pairwise preferences (T-REX) (Brown et al., 2019), or active queries using Ensemble Disagreement(EnsemDis), Ensemble Information Gain (EnsemInfo), Dropout Disagreement (DropDis), and Dropout Information Gain (DropInfo). N is set to 5 for Maze2D-Umaze, 10 for Maze2d-Medium and flow-merge-random, 15 for Halfcheetah. N is selected based on the size of the dimension of observation space and the complexity of the environment. Results are averages over three random seeds and are normalized so the performance of offline RL using the ground-truth rewards is 100.0 and performance of a random policy is 0.0. The results provide evidence that Ensembles are at least as good and often much better than random queries and that Dropout often fails to perform as well as random queries.

| | Query Aquisition Method | | | | |
| | Random | EnsemDis | EnsemInfo | DropDis | DropInfo |
|---|---|---|---|---|---|
| MAZE2D-UMAZE | 78.2 | **93.5** | 89.2 | 88.0 | 61.3 |
| MAZE2D-MEDIUM | 71.6 | **86.4** | 71.0 | 52.6 | 44.4 |
| HALFCHEETAH-RANDOM | 96.1 | **113.7** | 100.6 | 84.4 | 91.7 |
| FLOW-MERGE-RANDOM | **110.1** | 89.0 | 92.1 | 86.2 | 84.2 |
| KITCHEN-COMPLETE | 79.6 | 105.0 | **158.8** | 48.5 | 65.4 |
| HALFCHEETAH-MEDIUM-REPLAY | 98.8 | **100.0** | 99.7 | 96.2 | 94.0 |
| HALFCHEETAH-MEDIUM | 102.2 | **103.0** | 100.7 | 94.4 | 99.1 |
| HALFCHEETAH-MEDIUM-EXPERT | 111.4 | 91.0 | 112.8 | 86.0 | **113.9** |
| HALFCHEETAH-EXPERT | 104.3 | **176.4** | 124.6 | 86.0 | 129.3 |
| HOPPER-RANDOM | 81.9 | **86.7** | 64.7 | 72.0 | 75.1 |
| HOPPER-MEDIUM-REPLAY | 97.8 | 98.4 | **103.5** | 94.6 | 97.3 |
| HOPPER-MEDIUM | 90.4 | **93.6** | 91.7 | 89.4 | 92.0 |
| HOPPER-MEDIUM-EXPERT | 81.2 | 81.4 | 81.0 | **90.0** | 84.3 |
| HOPPER-EXPERT | 69.0 | 93.2 | **122.4** | 94.0 | 100.4 |
| KITCHEN-MIXED | 110.0 | **120.0** | 96.7 | 110.0 | 90.0 |
| KITCHEN-PARTIAL | 91.2 | 111.8 | 117.6 | **126.5** | 97.1 |

add 10 trajectories per round of active queries instead of 1 query per round. We found that increasing the number of queries was helpful due to the fact that MuJoCo tasks have higher dimensional observational spaces, which requires more data points to learn an accurate reward function. The learning curve is shown in Figure 4 (c). We found ensemble disagreement to be the best performing approach for the Halfcheetah task. We see a noticeable improvement as the number of active queries increases, however, 10 rounds of active queries is not sufficient to reach the same performance as the policy learned with the ground truth reward.

### 5.2.3 Flow Merge

The Flow Merge environment involves directing traffic such that the flow of traffic is maximized in a highway merging scenario. Results are shown in Table 2. Random queries achieved the best performance, but all query methods were able to recover near ground truth performance.

### 5.2.4 Robotic Manipulation

The Franka Kitchen domain involves interacting with various objects in the kitchen to reach a certain state configuration. The objects you can interact with include a water kettle, a light switch, a microwave, and cabinets. Our results in Table 1 show that setting the reward to all zero or a constant reward causes the performance to degrade significantly. Interestingly, the results in Table 2 show that OPRL using Ensemble Disagreement and Ensemble InfoGain are able to shape the reward in a way that actually leads to better performance than using the ground-truth reward for the task. In the appendix, we plot the learning curves and find that OPRL also converges to a better performance faster.

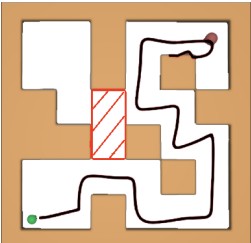

Figure 5: **Constrained Goal Navigation**. The highlighted yellow region represents a constraint region that the human prefers the agent to avoid while also traveling to the goal position shown in red. OPRL produces trajectories that match this preference by taking a more round-about, but more preferred, path to the goal.

### 5.3    New Offline Preference-Based Reward Learning Tasks

To explore the full benefits of learning a reward function from preferences, we propose and study several new tasks to highlight the need for a shaped reward function, rather than simply a goal indicator. To create these environments we adapted common gym environments including several that are in the D4RL set of benchmarks. The following section shows that OPRL can learn novel behaviors despite the offline data never containing successful or complete examples of these behaviors. We believe this is an exciting result that has not been previously demonstrated or recognized as possible in prior preference learning work.

#### 5.3.1    Maze Navigation with Constraint Region

We created a new variant of the Maze2d-Medium task where there is a constraint region in the middle of the maze that the supervisor does not want the robot to enter. Figure 5 shows this scenario where entering the highlighted yellow region is undesirable. After only 25 active queries using ensemble disagreement, we obtain the behavior shown in Figure 5 where the offline RL policy has learned to reach the goal while avoiding the constraint region.

#### 5.3.2    Open Maze Behaviors

Next, we took the D4RL Open Maze environment and dataset and used it to teach an agent to patrol the domain in counter-clockwise orbits, clockwise orbits, approximating a digital 2 shape, preferring to stay at the top, and approximating a "Z" shape. The open maze counter-clockwise orbiting uses human preferences provided by one of the authors by showing the human user trajectory snippets visualized as in 2 and querying preferences. The dataset only contains the agent moving to randomly chosen goal locations, thus this domain highlights the benefits of stitching together data from an offline database as well as the benefits of learning a shaped reward function rather than simply using a goal classifier to use as the reward function which has been used in prior work on offline RL work (Eysenbach et al., 2021). The original data distribution, samples of the preference queries, and the resulting learned policies are shown in figures 2 and 6. Our results involving the open maze behaviors in Figure 6 and our work on constrained maze in Figure 5 can be seen as a mixture of experts since the data consists of a mixture of demonstrations for a variety of starts and goals. OPRL can leverage diverse data to achieve different preferences (including ones not explicitly demonstrated), such as taking a longer rather than shorter path to a goal because of a specific user constraint preference in the medium maze, or various behaviors in the open maze.

#### 5.3.3    Open Ended CartPole Behaviors

We created an offline dataset of 1,000 random trajectories of length 200 in a modified CartPole domain where the trajectories only terminate at the end of 200 timesteps—this allows the pole to swing below the track and also allows the cart to move off the visible track to the left or right. Given this dataset of behavior agnostic data, we wanted to see whether we could optimize different policies, corresponding to different human preferences. We can also see this data from a random controller as highly suboptimal data for the following

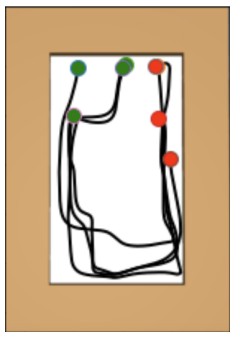 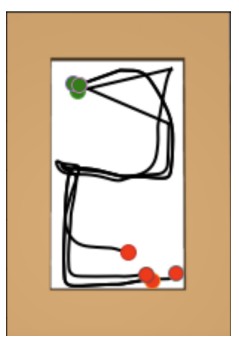 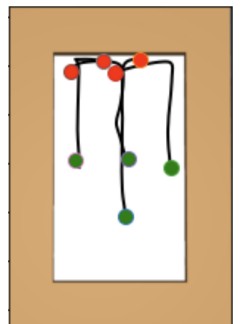 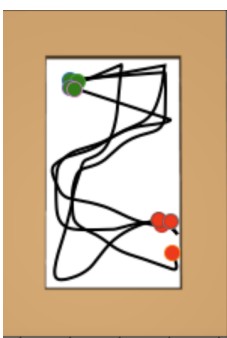

Figure 6: **Learned "Shape" Policies** adapt to particular user's preferences to learn counterclockwise orbits, a digital 2 shape, preferring to stay at the top, and a "Z" shape, respectively. Green circles represent random initial states and red circles represent the position of the agent at the end of a rollout.

Table 3: **Quantitative Results for Open Ended Cartpole Behaviors**: We evaluated the trained OPRL policies to calculate how often the pole was balanced ($\theta \leq \pm 24°$) and in view (cart position in [-2.4,2.4]) or in view and windmilling clockwise or counter-clockwise (by looking at the sign of the angular velocity). This table contains results showing the average (standard error of the mean) number of time steps in a 200-length trajectory that the different behaviors are exhibited in the randomly collected Offline Data (first row) and when rolling out the trained OPRL policy (second row).

|  | BALANCE | | WINDMILL (CW) | | WINDMILL (CCW) | |
|---|---|---|---|---|---|---|
|  | AVG. STEPS | SE | AVG. STEPS | SE | AVG. STEPS | SE |
| OFFLINE DATA | 23.07 | (0.37) | 71.39 | (2.01) | 72.12 | (2.06) |
| OPRL | **111.35** | (3.36) | **128.20** | (0.14) | **190.37** | (1.43) |

tasks. We optimized the following policies: *balance*, where the supervisor prefers the pole to be balanced upright and prefers the cart to stay in the middle of the track; *clockwise windmill*, where the supervisor prefers the cart to stay in the middle of the track but prefers the pole to swing around as fast as possible in the clockwise direction; and *counter-clockwise windmill*, which is identical to clockwise windmill except the preference is for the pole to swing in the counter-clockwise direction. OPRL is able to learn policies for all three behaviors. See the supplementary video for examples of the learned behaviors.

To quantitatively evaluate OPRL on these open-ended behaviors, we measure the average number of steps that OPRL policy is performing the task (e.g. balanced, windmill clockwise, windmill counter-clockwise) compared to the average number of steps these behavior appear in the original dataset as seen in Table 3. We see that OPRL is able to perform the task about 5 times more frequently for the balance task and around 2 times more frequently for the windmill tasks when compared to the original dataset. This gives evidence that OPRL is able to leverage offline data to optimize tasks not explicitly shown in the data.

### 5.3.4 Pilot User Study

We ran a small user study where we recruited 6 participants and asked them to give pairwise preference labels to teach an agent a goal location in Medium-Maze (Fig 3(b)) and a counter-clockwise orbiting behavior in OpenMaze (Sec 5.3.2). For Medium-Maze we quantitatively evaluated the performance of OPRL with respect to the GT reward from D4RL. Similar to our simulation results, we found that EnsemDis version of OPRL performed the best, achieving scores of 122.6 compared to GT performance of 134.6 and the remaining results are included in Table 10. All users were able to successfully use OPRL to teach an orbiting behavior nearly identical to Fig 6 (left) as shown in Fig 9 in the Appendix. In Table 10 we show results per user as well as results when aggregating all user preferences labels. We find that the combined data set (POOLED) achieved performance better than most of the individual users. We hypothesize this is because each user only answered 100 random queries which were used to create the pool for active learning. The POOLED results show the

benefit of having a large-enough active learning pool when performing offline preference learning. Overall, we found that users were able to provide preference labels that enable offline RL to generate behaviors aligned with human users.

## 6 Discussion and Future Work

Our goal is to learn a policy from user preferences without requiring access to a model, simulator, or interactions with the environment. Toward this goal, we propose OPRL: Offline Preference-based Reward Learning. We evaluate several different offline RL benchmarks and find that those that contain mainly random data or multi-task data are best suited for reward inference. While most prior work on reward learning has focused on learning from expert demonstrations, preference-based reward learning enables us to learn a user's intent even from random datasets by selecting informative sub-sequences of transitions and requesting pairwise preferences over these sub-sequences. Our results suggest that using ensemble disagreement as an acquisition function is the best choice since it leads to the best performance for 3 out of the 5 tasks and outperforms the random baseline for all 5 tasks. Additionally, using ensemble disagreement with only 10-15 preference queries we are able to learn policies with similar performance to policies trained with full access to tens of thousands of ground-truth reward samples. Finally, our results suggest there is surprising value in offline datasets even if the datasets do not explicitly contain data that matches a user's preferences or if the datasets contain suboptimal data. We are excited about the potential of offline reward and policy learning and believe that our work provides a stepping stone towards safer and more efficient autonomous systems that can better learn from and interact with humans. Preference-learning enables users to teach a system without needing to be able to directly control the system or demonstrate optimal behavior. We believe this makes preferences ideally suited for many tasks in as healthcare, finance, or robotics. Future work includes developing more complex and realistic datasets and benchmarks and developing offline RL algorithms and active query strategies that are well-suited for reward inference and offline RL.

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

## 7 Appendix

## A OPRL Performance with IQM

We present the results of methods using Interquartile Mean(IQM) instead of mean which was used in table 2. IQM is a statistic that is more robust to outliers than the mean and suggested for offline reinforcement learning by Agarwal et al. (2021). IQM is calculated by discarding the bottom and top 25% of the statistics and calculating the mean of the remaining 50%. For returns from 3 seeds, the IQM is calculated as $\dfrac{0.25R_1 + R_2 + 0.25R_3}{1.5}$ where $R_1 < R_2 < R_3$ are the returns sorted from lowest to highest.

Table 4: **OPRL Performance Using Reward Predicted After N Rounds of Querying using IQM**

|  | | QUERY ACQUISITION METHOD | | | |
|---|---|---|---|---|---|
|  | RANDOM | ENSEMDIS | ENSEMINFO | DROPDIS | DROPINFO |
| MAZE2D-UMAZE | 89.5 | **96.2** | 95.1 | 81.7 | 64.7 |
| MAZE2D-MEDIUM | 90.6 | **91.4** | 86.5 | 77.7 | 36.7 |
| HALFCHEETAH-RANDOM | 99.3 | **101.6** | 100.7 | 86.8 | 88.8 |
| KITCHEN-COMPLETE | 75.2 | **138.5** | 102.8 | 82.6 | 125.7 |
| HALFCHEETAH-MEDIUM-REPLAY | 99.6 | **100.3** | 99.7 | 96.7 | 93.5 |
| HALFCHEETAH-MEDIUM | 101.8 | **102.7** | 100.6 | 93.7 | 98.6 |
| HALFCHEETAH-MEDIUM-EXPERT | 63.7 | 65.0 | 76.3 | 68.4 | **108.6** |
| HALFCHEETAH-EXPERT | 81.8 | **127.3** | 66.4 | 56.3 | 87.0 |
| HOPPER-RANDOM | 72.8 | **74.1** | 69.2 | 55.2 | 57.4 |
| HOPPER-MEDIUM-REPLAY | 95.9 | 96.7 | **100.7** | 93.9 | 95.4 |
| HOPPER-MEDIUM | 91.4 | **91.9** | 89.1 | 87.4 | 89.7 |
| HOPPER-MEDIUM-EXPERT | 73.0 | 84.2 | **92.9** | 87.4 | 90.1 |
| HOPPER-EXPERT | 84.0 | 95.5 | 100.3 | **104.9** | 94.6 |
| KITCHEN-MIXED | 99.1 | **105.3** | 86.7 | 86.7 | 61.9 |
| KITCHEN-PARTIAL | 77.9 | **108.0** | 82.9 | 98.0 | 55.3 |

## B Sensitivity Analysis on Number of Queries

We present a sensitivity analysis on the number of initial queries and number of queries for ensemble disagreement in the halfcheetah-random-v2 environment.

Table 5: A sensitivity analysis is run on the half-cheetah-random-v2 environment. The default uses 50 initial queries and includes 10 additional queries per round. In this table, we vary the number of initial queries while keeping the additional number of queries per round fixed. The results are reported with 3 seeds and aggregated using the interquartile mean.

| NUMBER OF INITIAL QUERIES | 25 | 50 | 100 |
|---|---|---|---|
| RANDOM | 25.7 | 34.2 | 36.3 |
| ENSEMDIS | 26.4 | 41.6 | 36.9 |

We find that more initial queries improve performance for random queries. However, ensemble disagreement performs best with 50 initial queries. Ensemble disagreement outperforms random queries for all numbers of initial queries. The improvement from using ensemble disagreement vs. random queries is less significant for 25 and 100 queries. This is likely due to the fact that 25 queries are not sufficient to train an initial ensemble to estimate the disagreement and 100 queries by itself is sufficient to train a good reward model without additional active learning. 50 is likely strikes a good balance where we have enough to train a good initial ensemble of models while not diminishing the effect of using active queries.

Table 6: A sensitivity analysis is run on the half-cheetah-random-v2 environment. The default uses 50 initial queries and includes 10 additional queries per round. In this table, we vary the number of queries per round while keeping the initial number of queries fixed. The results are reported with 3 seeds and aggregated using the interquartile mean.

| Number of Queries per round | 5 | 10 | 20 |
|---|---|---|---|
| Random | 19.9 | 34.2 | 26.9 |
| EnsemDis | 31.1 | 41.6 | 32.5 |

We also vary the number of queries per round and found that ensemble disagreement outperforms random in all settings. The performance for ensemble disagreement is also fairly consistent across the different numbers of queries per round. We note that the more number of queries per round does not necessarily improve performance. However, this is likely due to the fact that other hyper-parameters like the number of iterations are tuned for 10 queries per round.

## C   Sensitivity Analysis on Number of Ensemble Models and Number of Dropout Samples

Table 7: A sensitivity analysis is run on the hopper-medium-v2 environment. The default uses 7 ensemble models. In this table, we vary the number of ensemble models for both ensemble disagreement and ensemble information gain. Results are aggregated with 3 seeds using interquartile mean.

| Number of Ensemble Models | 3 | 7 | 14 |
|---|---|---|---|
| EnsemDis | 1024.5 | 1116.1 | 1099.8 |
| EnsemInfo | 1088.8 | 1092.2 | 1128.6 |

For ensemble disagreement, we notice an improvement in performance from increasing the number of ensemble models from 3 to 7 and no improvement and a slight dip in performance from 7 to 14. For ensemble information gain, we notice that the performance improves slightly with more ensemble models. However, ensemble information gain seems to be less sensitive to the number of ensemble models compared to ensemble disagreement. This can also mean that ensemble information gain can be effective with fewer ensemble models. Although 14 ensemble models perform slightly better than the default in the case of ensemble disagreement, it does take significantly longer to train due to the additional number of ensemble models to train.

Table 8: A sensitivity analysis is run on the hopper-medium-v2 environment. The default uses 30 dropout samples. In this table, we vary the number of dropout samples for both dropout disagreement and dropout information gain. Results are aggregated with 3 seeds using interquartile mean.

| Number of Dropout Samples | 15 | 30 | 60 |
|---|---|---|---|
| DropDis | 1034.3 | 1054.3 | 1062.3 |
| DropInfo | 1064.3 | 1116.6 | 1087.0 |

For dropout disagreement, we notice that with more dropout samples, the performance improves. Dropout info gain on the other hand works best with 30 dropout samples and worse with 15 dropout samples. In both variants, increasing the number of dropout samples beyond 30 does not improve the performance significantly and even slightly degrades performance in the case of dropout information gain.

# D    Information Gain

As an alternative to disagreement, we also consider the expected information gain Cover (1999) between the reward function parameters $\theta$ and the outcome $Y$ of querying the human for a preference. We model our uncertainty using an approximate posterior $p(\theta \mid \mathcal{D})$, given by training an ensemble or dropout network on our offline dataset $\mathcal{D}$. Houlsby et al. (2011) show that the information gain of a potential query can be formulated as:

$$I(\theta; Y \mid \mathcal{D}) = H(Y \mid \mathcal{D}) - \mathbb{E}_{\theta \sim p(\theta \mid \mathcal{D})}[H(Y \mid \theta, \mathcal{D})]. \tag{3}$$

Intuitively, the information gain will be maximized when the first term is high, meaning that the overall model has high entropy, but the second term is low, meaning that each individual sample from the posterior has low entropy. This will happen when the individual hypotheses strongly disagree with each other and there is no clear majority. We approximate both terms in the information gain equation with samples obtained via ensemble or dropout.

Given a pair of trajectories $(\tau_A, \tau_B)$, let $Y = 0$ denote that the human prefers trajectory $A$ and $Y = 1$ denote that the human prefers trajectory $B$. Using the Bradley-Terry preference model Bradley & Terry (1952) we have that

$$P(Y = 0 \mid \theta, \mathcal{D}) = \frac{\exp(\beta R(\xi_A))}{\exp(\beta R(\xi_A)) + \exp(\beta R(\xi_B))} \tag{4}$$

where $R(\xi) = \sum_{(s,a) \in \tau_i} \hat{r}_\theta(s)$ and $P(Y = 1 \mid \theta, \mathcal{D}) = 1 - P(Y = 0 \mid \theta, \mathcal{D})$.

To perform queries using Information Gain, we evaluate a pool of potential trajectory pairs, evaluate the information gain for each pair and query the human for the pair that has the highest information gain.

$$\mathbb{E}_{\theta \sim p(\theta \mid \mathcal{D})}[H(Y \mid \theta, \mathcal{D})] \approx \frac{1}{M} \sum_{i=1}^{M} H(Y \mid \theta_i, \mathcal{D}), \tag{5}$$

where $\theta_i, i = 1, \ldots, M$ are $M$ approximate posterior samples from $p(\theta \mid \mathcal{D})$ obtained from each member of the ensemble or from Bayesian dropout sampling.

Because we only consider pairwise preference queries, computing the entropy corresponds to

$$H(Y|\theta_i, \mathcal{D}) = -\sum_{y=0}^{1} P(Y = y \mid \theta_i, \mathcal{D}) \log P(Y = y \mid \theta_i, \mathcal{D}) \tag{6}$$

To compute the first entropy term, we simply take the entropy of the average over the posterior as:

$$H(Y|\mathcal{D}) = -\sum_{y=0}^{1} \bar{p}_y \log \bar{p}_y \tag{7}$$

where $\bar{p}_y = \frac{1}{M} \sum_{i=1}^{M} P(Y = y \mid \theta_i, \mathcal{D})$.

# E    Evaluating Offline RL Benchmarks

We find that using average and zero masking are very competitive with using the true rewards on many of the benchmarks for D4RL and that these baselines often perform significantly better than a purely random policy.

This is a byproduct of offline reinforcement learning algorithms needing to learn in the presence of out-of-distribution actions. There are broadly two classes of methods towards solving the out-of-distribution problem. Behavioral cloning based methods like Advantage Weighted Regression used in our experiments, train only on actions observed in the dataset, which avoid OOD actions completely. Dynamic programming

(DP) methods, like BCQ, constrains the trained policy distribution to lie close to the behavior policy that generated the dataset. As a result of offline reinforcement learning algorithms constraining action to be similar to that of the static dataset, if the static dataset consists of exclusively expert actions, then the offline reinforcement learning algorithm will recover a policy that has expert like action regardless of the reward.

### E.1 Advantage-Weighted Regression

Consider Advantage-Weighted Regression with a constant reward function $r(s, a) = c$. The algorithm is simple. We have a replay buffer from which we sample transitions and estimate the value function via regression

$$V* \leftarrow \arg \min_{V} \mathbb{E}_{s,a \sim D} \left[ \| \mathcal{R}_{s,a}^{\mathcal{D}} - V(s) \| \right] \tag{8}$$

then the policy is updated via supervised learning:

$$\pi \leftarrow \arg \max_{\pi} \mathbb{E}_{s,a \sim D} \left[ \log \pi(a|s) \exp \left( \frac{1}{\beta} \left( \mathcal{R}_{s,a}^{\mathcal{D}} - V(s) \right) \right) \right] \tag{9}$$

If the rewards are all zero, then we have $V(s) = 0$, $\forall s \in \mathcal{S}$ and

$$\pi \leftarrow \arg \max_{\pi} \mathbb{E}_{s,a \sim D} \left[ \log \pi(a|s) \exp \left( \frac{1}{\beta} 0 \right) \right] \tag{10}$$

$$= \max_{\pi} \mathbb{E}_{s,a \sim D} \left[ \log \pi(a|s) \right]. \tag{11}$$

This is exactly the behavioral cloning loss which will learn to take the actions in the replay buffer. Thus, AWR is identical to BC when the rewards are all zero.

For non-zero, constant rewards, the advantage term will be zero (at least for deterministic tasks). If all trajectories from the state $s$ have the same length, then we will have zero advantage. However, if there are trajectories that terminate earlier than others, then AWR will put weights on these trajectories proportional to their length (for positive rewards $c > 0$) and inversely proportional to their length (for negative rewards $c < 0$). Thus, a terminal state will leak information and a good policy can be learned without an informative reward function.

## F  Comparison of AWR and CQL for OPRL

In Table 9 we show results for OPRL when using AWR Peng et al. (2019) for policy optimization and when using CQL Kumar et al. (2020) for policy optimization. Interestingly, our results suggest that when using CQL, using dropout to represent uncertainty performs better than using an ensemble. However, when using AWR for policy optimization, ensembles perform better. We hypothesize that different query mechanisms can have complex interactions with the actual policy learning algorithm. Better understanding how active queries and offline reward learning affects the performance of different offline RL algorithms is an interesting area of future work.

## G  Franka Kitchen Learning Curves

The FrankaKitchen domain involves interacting with various objects in the kitchen to reach a certain state configuration. The objects you can interact with include a water kettle, a light switch, a microwave, and cabinets. Our results in Table 1 show that setting the reward to all zero or a constant reward causes the performance to degrade significantly. However, the results in Table 2 show that OPRL using Ensemble Disagreement and Ensemble InfoGain are able to shape the reward in a way that leads to better performance than using the ground-truth reward for the task. In Figure 8, we plot the learning curves over time and find that OPRL also converges to a better performance faster. While surprising, these results support recent work showing that the shaping reward learned from pairwise preferences can boost the performance of RL, even when the agent has access to the ground truth reward function Memarian et al. (2021).

Table 9: **OPRL Performance Using Reward Predicted After N Rounds of Querying**: Offline RL performance using rewards predicted with T-REX using a static number of randomly selected pairwise preferences (T-REX), Ensemble Disagreement(EnsemDis), Ensemble Information Gain (EnsemInfo), Dropout Disagreement (DropDis), Dropout Information Gain (DropInfo) and Ground Truth Reward (GT). N is set to 5 for Maze2D-Umaze, 10 for Maze2d-Medium and flow-merge-random, 15 for Hopper and Halfcheetah. N is selected based on the size of the dimension of observation space and the complexity of the environment. Results are averages over three random seeds. Results are normalized such that the ground truth performance is 100.0 and the random policy performance is 0.0. We report the performance of OPRL using AWR Peng et al. (2019) and CQL Kumar et al. (2020) for policy optimization.

| AWR Peng et al. (2019) | Query Acquisition Method | | | | |
| --- | --- | --- | --- | --- | --- |
| | T-REX | EnsemDis | EnsemInfo | DropDis | DropInfo |
| maze2d-umaze | 78.2 | **93.5** | 89.2 | 88.0 | 61.3 |
| maze2d-medium | 71.6 | **86.4** | 71.0 | 52.6 | 44.4 |
| hopper | 69.0 | 77.5 | 72.8 | 87.6 | **90.6** |
| halfcheetah | 96.1 | **113.7** | 100.6 | 84.4 | 91.7 |
| flow-merge-random | **110.1** | 89.0 | 92.1 | 86.2 | 84.2 |
| kitchen-complete | 79.6 | 105.0 | **158.8** | 48.5 | 65.4 |
| CQL Kumar et al. (2020) | Query Acquisition Method | | | | |
| | T-REX | EnsemDis | EnsemInfo | DropDis | DropInfo |
| maze2d-umaze | 87.0 | 54.8 | 100.4 | **102.3** | 95.2 |
| maze2d-medium | 41.1 | 33.1 | **115.4** | 1.6 | 34.5 |
| hopper | 32.1 | 28.7 | 17.5 | 35.9 | **42.0** |
| halfcheetah | 68.4 | 75.9 | 76.3 | 75.0 | **79.6** |
| flow-merge-random | -13.6 | 44.3 | 103.2 | 69.6 | **114.0** |
| kitchen-complete | 85.7 | 100.0 | 71.4 | **114.3** | 71.4 |

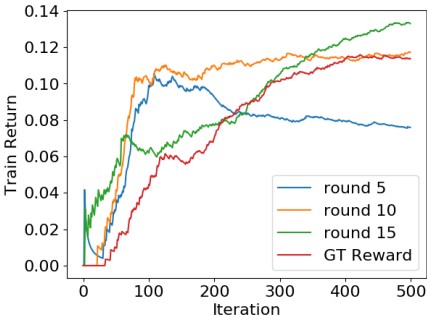

Figure 7: Kitchen-Complete-v1

Figure 8: Dropout disagreement after 15 rounds has similar performance to ground truth reward in Kitchen-complete

## H  Pairwise Preference Accuracy

Pairwise preference accuracy on a held-out set of trajectory pairs for each approach is provided in Table 11. Notably, when using randomly chosen queries (T-REX) the accuracy barely improves after round 5 in terms of ranking accuracy whereas all of our approaches continue to improve after round 5.

Table 10: **Quantitative Results for maze2d-medium-dense-v1 Using Human Labels**: To evaluate quantitative performance with human data, we collected 100 query labels from 6 different users in the maze2d-medium-dense-v1 environment. Due to the limitation that an ensemble of neural networks cannot be updated fast enough in between queries, we elected to create a pool of 100 queries per user. For the results below, we used 15 randomly selected preferences for Random and 15 active queries for all versions of OPRL. We notice that random queries slightly outperforms ensemble disagreement version of OPRL when trained on data from individual users but performs slightly worse in the pooled data which contained 600 queries. We attribute this to the observation that OPRL works better with larger datasets because the ability to pick informative queries becomes more important as the dataset gets larger. This also explains why ensemble disagreement version of OPRL consistently outperforms random queries on the original full dataset where the number of queries to choose from is order of magnitudes larger. This experiment demonstrates that humans are able to effectively answer queries that can be used to recover policies on par with GT performance.

| | \multicolumn{5}{c}{QUERY ACQUISITION METHOD} | |
| | RANDOM | ENSEMDIS | ENSEMINFO | DROPDIS | DROPINFO | GT |
| --- | --- | --- | --- | --- | --- | --- |
| POOLED | 122.5 | 122.6 | 103.6 | 106.2 | 106.9 | 134.6 |
| USER 1 | 118.60 | 114.98 | 119.72 | 106.97 | 116.85 | 134.6 |
| USER 2 | 119.81 | 114.40 | 114.48 | 93.68 | 107.32 | 134.6 |
| USER 3 | 118.68 | 114.39 | 114.51 | 112.97 | 107.87 | 134.6 |
| USER 4 | 123.21 | 121.97 | 120.54 | 112.07 | 115.07 | 134.6 |
| USER 5 | 116.29 | 118.87 | 109.30 | 96.84 | 101.09 | 134.6 |
| USER 6 | 121.88 | 121.89 | 122.48 | 112.05 | 109.23 | 134.6 |
| USER AVG | 119.75 | 117.25 | 116.84 | 105.76 | 109.57 | 134.6 |

## I   Quantitative Results with Human Labels

## J   Real human feedback experiments

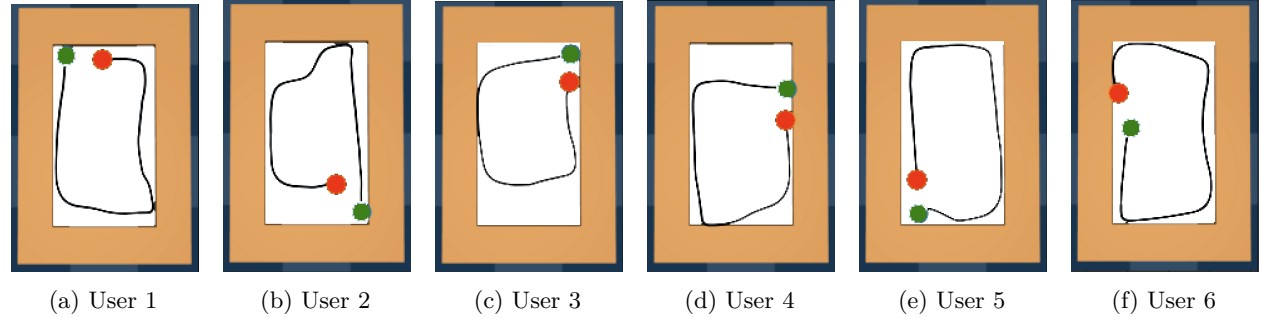

(a) User 1    (b) User 2    (c) User 3    (d) User 4    (e) User 5    (f) User 6

Figure 9: Counterclock wise orbit generated from human provided preferences

## K   Reward Model, Hyperparameter, And Dataset Details

During active reward learning, a total of 7 models are used in the ensemble variant. For the reward models, we used a standard MLP architecture using hidden sizes [512, 256, 128, 64, 32] with ReLU activations. For the dropout variant, we use 30 pass through of the model to calculate the posterior. Both the number of ensemble members and the number of pass-throughs were tuned to balance between speed and giving an accurate disagreement or information gain.

For policy learning with AWR, lower dimensional environments including Maze2D-Umaze, Maze2D-Medium, and Hopper are ran with 400 iterations. Higher dimensional environments including Halfcheetah, Flow-Merge-Random, and Kitchen-Complete are ran with 1000 iterations. The number of iterations are tuned so that the

policy converges before the maximum number of iterations. For CQL, policy learning rate is 1e-4, lagrange threshold is -1.0, min q weights is 5.0, min q version is 3, and policy eval start is 0. These numbers are suggested by the author of the CQL paper.

All results are collected and averaged across 3 random seeds. All models are trained on an Azure Standard NC24 Promo instance, with 24 vCPUs, 224 GiB of RAM and 4 x K80 GPU (2 Physical Cards). Datasets used to generate the customized behaviors (e.g. Constrained Goal Navigation, Orbit Policy, and Cartpole Windmill) will be released upon conference acceptance.

Table 11: Pairwise preference accuracy on held out set of trajectory pairs after 5 rounds and 10 rounds of queries respectively for different types of query acquisition functions.

|  | RANDOM | ENSEMDIS | ENSEMINFO | DROPDIS | DROPINFO |
|---|---|---|---|---|---|
| MAZE2D-UMAZE | 0.818, 0.867 | 0.817, 0.916 | 0.77, 0.862 | 0.874, 0.893 | 0.715, 0.755 |
| MAZE2D-MEDIUM | 0.646, 0.686 | 0.650, 0.783 | 0.638, 0.824 | 0.636, 0.795 | 0.692, 0.756 |
| HOPPER | 0.929, 0.922 | 0.943, 0.966 | 0.952, 0.966 | 0.946, 0.966 | 0.927, 0.960 |
| HALFCHEETAH | 0.814, 0.873 | 0.852, 0.942 | 0.866, 0.944 | 0.875, 0.927 | 0.833, 0.913 |
| FLOW-MERGE-RANDOM | 0.808, 0.841 | 0.829, 0.881 | 0.825, 0.861 | 0.846, 0.884 | 0.839, 0.893 |
| KITCHEN | 0.964, 0.978 | 0.981, 0.986 | 0.967, 0.984 | 0.973, 0.990 | 0.953, 0.986 |

## L  Table 1 Extended

Table 12: Offline RL performance on D4RL benchmark tasks comparing the performance with the true reward to performance using a constant reward equal to the average reward over the dataset (Avg) and zero rewards everywhere (Zero). Even when all the rewards are set to the average or to zero, many offline RL algorithms still perform surprisingly well. In this table, we present the experiments ran with BCQ. The degradation percentage is calculated as $\frac{\max(\text{GT}-\max(\text{AVG},\text{ZERO},\text{RANDOM}),0)}{|\text{GT}|} \times 100\%$.

| TASK | GT | AVG | ZERO | RANDOM | DEGRADATION % |
|---|---|---|---|---|---|
| FLOW-RING-RANDOM-V1 | 14.3 | -41.8 | -67.5 | -166.2 | 392.3 |
| FLOW-MERGE-RANDOM-V1 | 334 | 130.3 | 121.4 | 117 | 61.0 |
| MAZE2D-UMAZE | 104.2 | 36.5 | 41.8 | 49.5 | 52.5 |
| MAZE2D-MEDIUM | 106.6 | 23.3 | 28.5 | 44.8 | 58.0 |
| HALFCHEETAH-RANDOM | -0.6 | -1.5 | -1.9 | -285.8 | 133.9 |
| HALFCHEETAH-MEDIUM-REPLAY | 3400.0 | 2659.3 | 3185.0 | -285.8 | 6.3 |
| HALFCHEETAH-MEDIUM | 3057.7 | 2219.1 | 2651.7 | -285.8 | 13.3 |
| HALFCHEETAH-MEDIUM-EXPERT | 10905.8 | 11146.9 | 11030.3 | -285.8 | 0.0 |
| HALFCHEETAH-EXPERT | 63.6 | 115.8 | -32.4 | -285.8 | 0.0 |
| HOPPER-RANDOM | 199.8 | 186.0 | 199.4 | 18.3 | 0.2 |
| HOPPER-MEDIUM-REPLAY | 1188.1 | 1082.4 | 908.3 | 18.3 | 8.9 |
| HOPPER-MEDIUM | 922.3 | 745.4 | 842.2 | 18.3 | 8.7 |
| HOPPER-MEDIUM-EXPERT | 343.3 | 225.7 | 202.0 | 18.3 | 34.3 |
| HOPPER-EXPERT | 241.5 | 268.1 | 343.3 | 18.3 | 0.0 |
| KITCHEN-COMPLETE | 0.07 | 0.11 | 0.04 | 0.0 | 0.0 |
| KITCHEN-MIXED | 0.08 | 0.07 | 0.02 | 0.0 | 12.5 |
| KITCHEN-PARTIAL | 0.16 | 0.17 | 0.13 | 0.0 | 0.0 |

Table 13: Offline RL performance on D4RL benchmark tasks comparing the performance with the true reward to performance using a constant reward equal to the average reward over the dataset (Avg) and zero rewards everywhere (Zero). Even when all the rewards are set to the average or to zero, many offline RL algorithms still perform surprisingly well. In this table, we present the experiments ran with BEAR. The degradation percentage is calculated as $\frac{\max(\text{GT}-\max(\text{AVG},\text{ZERO},\text{RANDOM}),0)}{|\text{GT}|} \times 100\%$

| TASK | GT | AVG | ZERO | RANDOM | DEGRADATION % |
|---|---|---|---|---|---|
| FLOW-RING-RANDOM-V1 | 13.4 | 18.2 | 11.6 | -166.2 | 0.0 |
| FLOW-MERGE-RANDOM-V1 | 75.9 | 62.1 | 68.3 | 117.0 | 0.0 |
| MAZE2D-UMAZE | 66.6 | 44.8 | 59.7 | 49.5 | 10.3 |
| MAZE2D-MEDIUM | 18.6 | 29.8 | 22.5 | 44.8 | 0.0 |
| HALFCHEETAH-RANDOM | 0.8 | -0.9 | -0.5 | -285.8 | 160.5 |
| HALFCHEETAH-MEDIUM-REPLAY | 4997.5 | 3919.6 | 4282.7 | -285.8 | 14.3 |
| HALFCHEETAH-MEDIUM | 5260.9 | 4910.3 | 4941.4 | -285.8 | 6.1 |
| HALFCHEETAH-MEDIUM-EXPERT | 5346.9 | 5033.4 | 4709.1 | -285.8 | 5.9 |
| HALFCHEETAH-EXPERT | 11329.1 | 11268.9 | 10627.6 | -285.8 | 0.5 |
| HOPPER-RANDOM | 221.9 | 215.6 | 86.3 | 18.3 | 2.9 |
| HOPPER-MEDIUM-REPLAY | 1261.9 | 659.0 | 1311.2 | 18.3 | 0.0 |
| HOPPER-MEDIUM | 1564.3 | 1749.3 | 1601.5 | 18.3 | 0.0 |
| HOPPER-MEDIUM-EXPERT | 2269.7 | 2995.2 | 1387.9 | 18.3 | 0.0 |
| HOPPER-EXPERT | 2110.6 | 2925.9 | 2826.3 | 18.3 | 0.0 |
| KITCHEN-COMPLETE | 0.8 | 0.7 | 0.1 | 0.0 | 12.0 |
| KITCHEN-MIXED | 0.7 | 0.6 | 0.2 | 0.0 | 14.3 |
| KITCHEN-PARTIAL | 0.2 | 0.3 | 0.1 | 0 | 0.0 |

Table 14: Offline RL performance on D4RL benchmark tasks comparing the performance with the true reward to performance using a constant reward equal to the average reward over the dataset (Avg) and zero rewards everywhere (Zero). Even when all the rewards are set to the average or to zero, many offline RL algorithms still perform surprisingly well. In this table, we present the experiments ran with CQL. The degradation percentage is calculated as $\frac{\max(\text{GT}-\max(\text{AVG},\text{ZERO},\text{RANDOM}),0)}{|\text{GT}|} \times 100\%$.

| TASK | GT | AVG | ZERO | RANDOM | DEGRADATION % |
|---|---|---|---|---|---|
| FLOW-RING-RANDOM-V1 | 13.4 | -49.5 | -23.3 | -166.2 | 274.1 |
| FLOW-MERGE-RANDOM-V1 | 156.1 | 98.1 | 50.5 | 117.0 | 25.0 |
| MAZE2D-UMAZE | 83.4 | 46.0 | 38.2 | 49.5 | 40.7 |
| MAZE2D-MEDIUM | 107.9 | 34.3 | 19.6 | 44.8 | 58.5 |
| HALFCHEETAH-RANDOM | 3140.7 | -281.5 | -158.5 | -285.8 | 105.0 |
| HALFCHEETAH-MEDIUM-REPLAY | 5602.9 | 1145.4 | -512.2 | -285.8 | 79.6 |
| HALFCHEETAH-MEDIUM | 5546.9 | 5130.2 | 5037.2 | -285.8 | 7.5 |
| HALFCHEETAH-MEDIUM-EXPERT | 3544.6 | 4372.7 | 5505.3 | -285.8 | 0.0 |
| HALFCHEETAH-EXPERT | 258.2 | 45.5 | 3546.6 | -285.8 | 0.0 |
| HOPPER-RANDOM | 968.9 | 512.0 | 11.0 | 18.3 | 47.2 |
| HOPPER-MEDIUM-REPLAY | 2484.2 | 1970.3 | 563.7 | 18.3 | 20.7 |
| HOPPER-MEDIUM | 1709.9 | 2205.4 | 2098.7 | 18.3 | 0.0 |
| HOPPER-MEDIUM-EXPERT | 1156 | 1136.9 | 1210.9 | 18.3 | 0.0 |
| HOPPER-EXPERT | 919.1 | 904.1 | 2092.7 | 18.3 | 0.0 |
| KITCHEN-COMPLETE | 0.8 | 0.7 | 0.6 | 0.0 | 14.3 |
| KITCHEN-MIXED | 0.6 | 0.4 | 0.8 | 0.0 | 0.0 |
| KITCHEN-PARTIAL | 0.3 | 0.7 | 0.4 | 0.0 | 0.0 |

