# OpenReview forum: "Benchmarks and Algorithms for Offline Preference-Based Reward Learning"
_TMLR — Accepted by TMLR_

### Review · Reviewer_Ndnj · 2022-10-05

**Summary Of Contributions:**

This paper proposes to leverage human preference and active queries in offline RL without ground-truth reward labels. The work is built upon the previous work T-REX where the reward function is learned with pairwise preference provided by humans but instead of using randomly selected pairs and online rollouts in T-REX, the method picks pairs of trajectories using active queries with uncertainty estimation on static datasets. The authors show that the approach with various uncertainty estimation techniques for active queries can perform well in many offline RL benchmarks and sometimes matches or even outperforms the results of offline RL with true rewards. In the end, the authors also provided additional benchmarks where the proposed method can learn behaviors that are not shown in the offline dataset.

**Requested Changes:**

1. Clarify the importance of using active queries and compare them to T-REX with more randomly sampled pairwise preferences per round.

2. Normalize the results w.r.t. expert scores and tune the backbone offline RL algorithms to make sure the results can match the original paper numbers.

3. Compare with other reward learning/imitation learning methods such as DemoDICE, IQ-Learn and etc.

**Strengths And Weaknesses:**

**Strengths:**

1. The method investigates how preference-based learning can work in offline RL settings where rewards are not present, which is an important and practical problem for the community.

2. The empirical results of the paper clearly show that the method can perform relatively well in many scenarios compared to the approach with ground-truth rewards, which is surprising yet interesting.

3. The paper is well-written and easy to follow.

**Weaknesses**:

1. I think the method of the approach is a bit incremental. It seems a bit derivative from the T-REX paper. Though the authors added the active query part with various uncertainty estimation techniques, the improvement over T-REX is a bit marginal and highly task/algorithm-dependent. I also wonder if the performance of T-REX could drastically improve with a large number of randomly selected pairwise preferences.

2. It is a bit strange that the results are normalized w.r.t. the policy learned by offline RL with ground-truth rewards. The authors should use the expert score to make it clear how these approaches are performing. Moreover, The results of offline RL with ground-truth rewards are significantly worse than the numbers shown in the original papers, especially CQL. I think the authors should tune the backbone policy optimization algorithms better and make sure the results can match the results in the original papers. Otherwise, the results shown here could be misleading.

3. I think the authors should also compare the method to other reward learning/imitation learning methods such as DemoDICE, IQ-Learn etc.

---

> ### Author Response · Authors · 2022-10-21
> **Response**
>
> > I think the method of the approach is a bit incremental. It seems a bit derivative from the T-REX paper.
>
> We agree that our approach is related to T-REX and other preference learning algorithms. However, we argue that our paper should not be judged relative to perceived novelty in terms of algorithm design. Rather, we emphasize the following contributions and distinctions from prior work. (1) We study preference-based RL in the offline settings, (2) we evaluate different methods for active query selection and uncertainty approximation, both of which T-REX did not consider—T-REX assumes a fixed set of preference labels, (3) We evaluate common offline-RL benchmarks, show that many are not well-suited for evaluating good reward learning, and propose several new benchmarks that require learning a good reward function to perform well and are thus well suited for studying methods like OPAL.
>
> >Though the authors added the active query part with various uncertainty estimation techniques, the improvement over T-REX is a bit marginal and highly task/algorithm-dependent.
>
> We would like to emphasize that our experimental results give evdience that ensemble disagreement performs well across a variety of benchmarks. In particular, we find that OPAL ensemidis performs 12.5% better, on average, than random preference queries and has at least a marginal gain over random preference queries on 14 out of 16 domains.
> The more data sampled, the better Random sampling will do, and eventually it should do just as well as active sampling. Thus, rather than focusing on the performance in the limit, we are interested in performance given small numbers of active vs. random queries since we want this to be applicable to human-in-the-loop training where we can't ask large numbers of queries.
>
> >I also wonder if the performance of T-REX could drastically improve with a large number of randomly selected pairwise preferences.
>
> Great question. As with any active learning method, the benefit of actively querying for data will diminish with large numbers of queries—as the number of queries increases the difference in performance between random query selection and active query selection will decrease. The main benefit of active learning is in low-sample regimes which we study in this paper. However, we agree that a senstivity analysis is valuable and are currently running a new experiment that we include in the appendix that compares the number of initial queries used, and the number of queries per round. We will add these results as soon as they are completed.
>
> >It is a bit strange that the results are normalized w.r.t. the policy learned by offline RL with ground-truth rewards. The authors should use the expert score to make it clear how these approaches are performing.
>
> We are unsure what the reviewer means by expert scores. Because we do not learn from demonstrations, there are no expert scores per se. We did find these min and max numbers provided by D4RL (https://github.com/Farama-Foundation/D4RL/blob/master/d4rl/infos.py). However, it is unclear how some of theses scores were determined (the white paper says many were just estimates) and many of these scores use online RL or behavioral cloning on expert demos to determine “expert” performance. We do not think these notions of expert make sense in our setting where there is no access to expert demonstrations nor any access to the dynamics of the environment, nor the true reward function. Thus, we chose to normalize with respect to the performance of an offline RL policy that uses exactly the same hyper parameters as OPAL but differs in that it has access to the ground-truth reward, whereas OPAL must first estimate this reward function from active preference queries. However, we would be happy to report scores with respect to the min and max values in the above link if that is what the reviewer was referring to.

---

> > ### Author Response · Authors · 2022-10-21
> > **Response Continued**
> >
> > >The results of offline RL with ground-truth rewards are significantly worse than the numbers shown in the original papers, especially CQL. I think the authors should tune the backbone policy optimization algorithms better and make sure the results can match the results in the original papers.
> >
> > We attempted as much as possible to follow the guidance for hyperparameters provided in the D4RL paper and in the offline RL companion repo, but found the reported D4RL performance on several of the tasks very hard to replicate even with further parameter tuning. For example we used the author's implementation of CQL (https://github.com/aviralkumar2907/CQL/blob/master/README.md). We agree that some of the results are lower than those reported in D4RL; however, some of our results are also much higher. Despite these differences, we would like to emphasize that we are not proposing a state-of-the-art offline RL algorithm. Rather, we seek to study the offline apprenticeship learning setting. In this case we fix the RL algorithm and only change the reward. This allows us to measure the difference between offline RL with the ground-truth reward versus offline RL with the inferred reward. Thus, even though there are likely some differences in hyperparameters between our results and those in D4RL, we do not see this as a fundamental problem since we use the same hyperparameters when evaluating different reward learning variants and use the same hyperparameters for RL with the true rewards, allowing fair comparison and allowing us to isolate the effect of the learned reward.
> >
> > > I think the authors should also compare the method to other reward learning/imitation learning methods such as DemoDICE, IQ-Learn etc.
> >
> > We appreciate the suggestion to add further baselines, but do not think that we can present a good-faith, apples-to-apples comparison between our method and DemoDICE and IQ-learn. The main reason is that both DemoDICE and IQ-learn assume access to expert demonstrations. However, in our problem setup we just have access to an offline dataset that may come from a random policy or policies that are suboptimal or optimize different rewards (e.g. maze). Thus, we feel that any results comparing DemoDICE or IQ-learn on these datasets would be misleading since these algorithms are not designed to work in the preferences-based RL setting. Because our method is an active preference-based RL method and as such we think that the current comparison we include in the paper (evaluating different query methods and methods for representing uncertainty) is the best way to evaluate OPAL. In summary, we think it would be unfair to run these other algorithms in a setting where there are no expert demos, where we would expect them to fail. Alternatively, we do not expect OPAL to perform as well as methods that are provided expert demonstrations since OPAL does not have access to demonstrations that are known to come from an expert and must learn from qualitative preferences over trajectories of highly variable quality.

---

> ### Author Response · Authors · 2022-10-27
> **Response**
>
> >Clarify the importance of using active queries and compare them to T-REX with more randomly sampled pairwise preferences per round.
>
> We added a sensitivity analysis on the number of initial queries and number of queries for ensemble disagreement in the halfcheetah-random-v2 environment in section A of the appendix.
>
> The results are reported with 3 seeds and aggregated using the interquantile mean. We find that more initial queries improve performance for random queries. However, ensemble disagreement performs best with 50 initial queries. Ensemble disagreement outperforms random queries for all numbers of initial queries. The improvement from using ensemble disagreement vs. random queries is less significant for 25 and 100 queries. This is likely due to the fact that 25 queries are not sufficient to train an initial ensemble to estimate the disagreement and 100 queries by itself is sufficient to train a good reward model without additional active learning. 50 is likely strikes a good balance where we have enough to train a good initial ensemble of models while not diminishing the effect of using active queries.
>
> We also vary the number of queries per round and found that ensemble disagreement outperforms random in all settings. The performance for ensemble disagreement is also fairly consistent across the different numbers of queries per round. We note that the more number of queries per round does not necessarily improve performance. However, this is likely due to the fact that other hyper-parameters like the number of iterations are tuned for 10 queries per round.
>
> We are also finalizing a sensitivity analysis on the number of ensemble models and the number of samples for dropout that we believe will address this concern and also provide more insights for future research.

---

### Review · Reviewer_2gDE · 2022-10-06

**Summary Of Contributions:**

The paper describes an approach for learning a reward function based on human preferences. This is an approach with a long history in the literature, and this paper focuses specifically on learning a reward function assuming access to an offline dataset of interaction, and sampling episode segments from the offline dataset, rather than doing online interaction.

In the process of doing so, they consider the following axes:

* How should paired trajectories be sampled? Different active learning techniques used.
* How should uncertainty be estimated for the active learning techniques? Compared ensembles to Bayesian dropout
* What environments are good ones to evaluate learned reward functions? Compared no-reward baselines to check which environments within D4RL seem more benefit from a good learned reward.

The work is primarily done using the ground truth reward to answer queries, with the last section focusing on more qualitative tasks with human feedback.

**Broader Impact Concerns:**

No concerns.

**Requested Changes:**

The caption for Figure 2 appears to be repeated 3 times in the submission on page 3, please clean this up. (critical)

"identify a subset environments" -> subset of environments (nitpick)

I do not think it is worth claiming that this is the first paper to evaluate dropout disagreement and ensembles for an info gain acquistion function. Although I am not sure of a specific learning from human preferences paper that does so, approaches like this are quite common for active learning in general, and if restricted to human preference learning, the claim is specific enough to not really be interesting to make. (decently important)

It would be good if the authors could add more algorithmic details on how they efficiently identify the best pair of trajectories to compare. A naive approach for IDing the best pair would be O(N^2) in dataset size, checking every pair, and I assume the authors did not do this. (decently important)

The paths in Figure 5 in black are hard to see against the blue background. The red color from Figure 5 would be better here. (nitpick)

**Strengths And Weaknesses:**

The work is reasonably careful on comparing different methods for surfacing queries to request labels for, eventually finding that when AWR is used, an ensemble disagreement approach works best. The authors also identify offline datasets where a reward-agnostic approach (all zero or all constant reward) performs well - these datasets are primarily datasets generated by expert policies, and are ones where a learned reward function is not too helpful for the underlying RL algorithm because offline RL algorithms tend to act BC-like when they do not have reward information.

The technical work seems solid and it refers to appropriate prior work for learning from human preferences - as noted in the paper, most of the approaches are known but this should be of interest for others, especially on which benchmarks are suitable for benchmarking learned reward functions.

The primary changes needed are general cleanup, see next section.

---

> ### Author Response · Authors · 2022-10-21
> **Response**
>
> We have fixed the typos.
>
> > I do not think it is worth claiming that this is the first paper to evaluate dropout disagreement and ensembles for an info gain acquistion function. Although I am not sure of a specific learning from human preferences paper that does so, approaches like this are quite common for active learning in general, and if restricted to human preference learning, the claim is specific enough to not really be interesting to make.
>
> We agree and have removed this claim from the paragraph before section 4.1.
>
> >It would be good if the authors could add more algorithmic details on how they efficiently identify the best pair of trajectories to compare. A naive approach for IDing the best pair would be O(N^2) in dataset size, checking every pair, and I assume the authors did not do this.
>
> Thank you for this suggestion. We have added a paragraph in 4.2 called “Searching the Offline Dataset for Informative Queries” where we discuss efficiency. Note that we did not find it necessary to parallelize the computation of information gain and ensemble disagreement for our experiments, but we did take advantage of GPU parallelization to significantly speed up our search.
>
> >The paths in Figure 5 in black are hard to see against the blue background. The red color from Figure 5 would be better here. (nitpick)
>
> Thank you for the suggestion. We will update these figures to make the paths easier to see.

---

> > ### Comment · Reviewer_2gDE · 2022-10-27
> > **Reply**
> >
> > Thanks for the updates.
> >
> > Re the note as an any-time algorithm: do the authors treat it as an anytime algorithm within the paper via random sampling, or is the informative query done completely and in parallel?

---

> > > ### Author Response · Authors · 2022-10-28
> > > **Response**
> > >
> > > In our implementation, we utilize random sampling and GPU parallelization but don’t implement it as an anytime algorithm since the quality of active queries is the emphasis. However, we think this is an interesting area for future work and valuable in cases where there may be tight time constraints.

---

### Review · Reviewer_yMyt · 2022-10-07

**Summary Of Contributions:**

This paper proposes a method for leveraging offline RL data for preference-based learning. That is, the offline trajectories can be used to query user preferences and learn a reward function for them. Their proposed method incorporates a measure of reward uncertainty, query selection, and policy optimization. The main contributions of the paper are in establishing offline preference-based learning as an interesting problem worty of study, evaluating a few natural baselines, and determining whether existing offline RL datasets prove challening enough for this type of problem.

**Broader Impact Concerns:**

No concerns from me on this front.

**Requested Changes:**

## Major changes / questions
1. In the abstract, in the sentences "To test our approach, we first ... which allow for more open-ended behaviors.", it's not clear whether you're talking about preference learning or reward learning in general. I believe you're referring to preference-based, but worth making a bit more explicit.
1. In the second-to-last paragraph in the second page, when discussing benchmarks that are "ill-suited for reward learning", could one consider this more indicative of the success of existing offline RL algorithms, as opposed to failures in the benchmarks?
1. The caption of Figure 2 is repeated 3 times.
1. The lines right before Section 4.1 saying "We are the first to evaluate Dropout Disagreement..." seems like a weird and unnecessary phrase to include, given that the two components listed are what make up OPAL; so the statement is akin to saying: "We are the first to evaluate the method we are introducing"... I'd suggest removing it.
1. In FIgure 4 it is not clear what "rounds" represents.
1. In Figure 4, how many seeds were run? What do the shaded areas represent?
1. In section 5.2 the authors specify how many preference queries (initial + active) were used for the experiments in each domain. How were these selected? How sensitive is your method to this choice? It would be nice to provide some type of sensitivity analysis here, as this is an important decision. What is also lacking is some kind of insight/suggestion for future researchers. If someone uses a different offline dataset with a (possibly) different method, how would they know how many queries are sufficient?
1. In Table 2, since all results are normalized, you should report more robust statistics as suggested in [2](https://proceedings.neurips.cc/paper/2021/hash/f514cec81cb148559cf475e7426eed5e-Abstract.html) (namely, Interquantile Mean)
1. The trajectories in Figures 5 and 6 are hard to see, I would suggest removing the dark blue background.
1. In Figure 6, it would be good to overlay multiple trajectories (instead of just one) so we can see the consistency of the method.
1. In Appendix H it is specified that 7 models are used in the ensemble variant, and 30 pass-throughs used to calculate the posterior. How sensitive is the method to these choices? It would be good to provide a sensitivity analysis.


## Minor / cosmetic changes:
1. In abstract, should read "using a trivial reward function results **in** good policy performance"
1. In page 2 in the second, third, and last paragraphs you refer to "Figure 1" but it should be "Figure 2"
1. Page 2, second paragraph: "components of our framework **are** not novel by **themselves**."
1. Another paper that may be useful to discuss in related work is [1](https://arxiv.org/abs/1907.13411).
1. In the last line of page 4: "of demonstrations for a vari**e**ty of tasks."
1. In the second-to-last sentence before section 5.1 it would read better as "**Finally, we** propose and..."
1. In Section 5.2.2 it says that after 10 rounds of querying results are "on par with the policy learned with the ground truth reward", but it's not quite so, it seems a fair bit lower, according to Figure 4.
1. In the third line of section 5.3.2 it should read "The open maze **counter-clockwise**..."
1. In the line above equation (4) in the appendix, it should say "**Using** the Bradley-Terry preference..."

[1] [Castro, Li, & Zhang, 2019; Inverse Reinforcement Learning with Multiple Ranked Experts.](https://arxiv.org/abs/1907.13411)

[2] [Agarwal, Schwarzer, Castro, Courville, & Bellemare, 2021; Deep reinforcement learning at the edge of the statistical precipice.](https://proceedings.neurips.cc/paper/2021/hash/f514cec81cb148559cf475e7426eed5e-Abstract.html)

**Strengths And Weaknesses:**

Overall I think this paper is a nice contribution, and suggests an interesting problem which, as far as I know, hasn't been introduced before. I have a few (fixable) concerns which will make the paper stronger.

## Strengths
1. I like the idea of preference-based learning from offline datasets, and I think it shows promise for future research.
1. I think the authors did a good job in evaluating some of the D4RL environments to determine which of these are well-suited for preference-based learning.
1. The derivation in Appendix B is quite nice!

## Weaknesses
1. One aspect that recurs throughout the paper is the notion of estimating/representing uncertainty. While I can imagine reasons why this is important, I feel the paper could do a better job at justifying _why_ it is important to estimate/represent uncertainty. In its current form, it seems to take this for granted.
1. Another recurring issue is that a number of design decisions for the proposed algorithm (number of queries, number of ensemble models, etc.) are specified but not well-justified. It would be good to provide sensitivity analyses for these, as it will help future researchers/users be better equipped for dealing with these types of problems methods.

---

> ### Author Response · Authors · 2022-10-21
> **Response**
>
> Thank you for your thoughtful and thorough review. We have fixed the noted cosmetic changes and typo/wording changes. Major changes to the draft are shown in blue font. Below we address the remaining questions and concerns.
>
> > One aspect that recurs throughout the paper is the notion of estimating/representing uncertainty. While I can imagine reasons why this is important, I feel the paper could do a better job at justifying why it is important to estimate/represent uncertainty. In its current form, it seems to take this for granted.
>
> Thank you for the suggestion. We agree that this is an important point that shouldn’t be taken for granted. We have clarified in contribution 3 that the uncertainty estimation is used when selecting queries during active learning. Without any notion of uncertainty, we cannot perform active learning to determine which queries would be best to ask in order to reduce uncertainty and become more confident in the correct reward function. We have additionally added a sentence clarifying this to the paragraph right before section 4.1. Please let us know if there are other places you think this could be better clarified.
>
> > Another recurring issue is that a number of design decisions for the proposed algorithm (number of queries, number of ensemble models, etc.) are specified but not well-justified. It would be good to provide sensitivity analyses for these, as it will help future researchers/users be better equipped for dealing with these types of problems methods.
>
> We agree. We are currently finalizing a sensitivity analysis on several of our design choices for OPAL in terms of initial and per round queries which we will add to the appendix.
>
> >In the abstract, in the sentences "To test our approach, we first ... which allow for more open-ended behaviors.", it's not clear whether you're talking about preference learning or reward learning in general. I believe you're referring to preference-based, but worth making a bit more explicit.
>
> We have clarified this in the abstract. Our analysis of offline RL benchmarks shows that many are ill-suited for evaluating any kind of reward learning. We also clarified that our new benchmarks are specifically designed for offline preference-based reward learning.
>
> >In the second-to-last paragraph in the second page, when discussing benchmarks that are "ill-suited for reward learning", could one consider this more indicative of the success of existing offline RL algorithms, as opposed to failures in the benchmarks?
>
> This is an interesting point. The reason that we say they are ill-suited for reward learning evaluation is that the choice of reward function does not seem to matter. Indeed, we were surprised that the all-zero reward function or a constant average leads to performance as good as using the ground-truth reward function. While we agree that current off-line RL algorithms are successful, the success of these algorithms when given absolute no reward signal (all zeros) points to a problem with the data. Many domains contain mostly or all expert data, thus offline-rl amounts to learning how to best imitate this data, and the reward function doesn’t play much of a role. By contrast, we are focused on learning reward functions. Thus, we seek tasks where the choice of reward function has a strong effect on performance, otherwise we cannot tell whether one method for learning rewards is better or worse than another. We have added clarification to this paragraph.
>
> >The lines right before Section 4.1 saying "We are the first to evaluate Dropout Disagreement..." seems like a weird and unnecessary phrase to include, given that the two components listed are what make up OPAL; so the statement is akin to saying: "We are the first to evaluate the method we are introducing"... I'd suggest removing it.
>
> Good point. We have removed this sentence as suggested.
>
> >In FIgure 4 it is not clear what "rounds" represents.
>
> We have clarified that these are rounds of active queries and have clarified the number of active queries used per round.
>
> >In Figure 4, how many seeds were run? What do the shaded areas represent?
>
> Three seeds. Shaded area represents standard deviation.
>
> >In section 5.2 the authors specify how many preference queries (initial + active) were used for the experiments in each domain. How were these selected? How sensitive is your method to this choice? It would be nice to provide some type of sensitivity analysis here, as this is an important decision. What is also lacking is some kind of insight/suggestion for future researchers. If someone uses a different offline dataset with a (possibly) different method, how would they know how many queries are sufficient?
>
> We are currently finalizing a sensitivity analysis on numbers of preferences (initial + active) that we believe will address this concern and also provide more insights for future research.

---

> > ### Author Response · Authors · 2022-10-21
> > **Response continued**
> >
> > >In Table 2, since all results are normalized, you should report more robust statistics as suggested in 2 (namely, Interquantile Mean)
> >
> > Thank you for the suggestion. We only ran three seeds so this would amount to reporting the median performance, right? We will update the tables accordingly.
> >
> > >The trajectories in Figures 5 and 6 are hard to see, I would suggest removing the dark blue background.
> >
> > Thank you for the suggestion. We will update the figures as recommended.
> >
> > >In Figure 6, it would be good to overlay multiple trajectories (instead of just one) so we can see the consistency of the method.
> >
> > Thank you for the suggestion. We will regenerate these figures with multiple rollouts.
> >
> > > In Appendix H it is specified that 7 models are used in the ensemble variant, and 30 pass-throughs used to calculate the posterior. How sensitive is the method to these choices? It would be good to provide a sensitivity analysis.
> >
> > Thank you for this suggestion. We limited the number of ensembles to 7 with 30 dropout passes to enable us to have a robust approximation of the posterior distribution over preferences while making things computationally feasible to run large scale experiments. We are currently running a sensitivity analysis on the effect of using fewer ensemble members and fewer passes and will add these to the appendix.
> >
> > >Another paper that may be useful to discuss in related work is 1.
> >
> > We have added this to our related work section.
> >
> > >In Section 5.2.2 it says that after 10 rounds of querying results are "on par with the policy learned with the ground truth reward", but it's not quite so, it seems a fair bit lower, according to Figure 4.
> >
> > Thank you for pointing this out. We agree and have changed the wording in 5.2.2.

---

> > > ### Comment · Reviewer_yMyt · 2022-10-26
> > > **Response to authors**
> > >
> > > Thank you for your responses! I look forward to seeing the sensitivity analyses in the appendix. I have one response to a question you posed in your response:
> > >
> > > > We only ran three seeds so this would amount to reporting the median performance, right?
> > >
> > > No, IQM is a more robust statistic than median, and was developed precisely for situations with few seeds.

---

> > > > ### Author Response · Authors · 2022-10-27
> > > > **Response**
> > > >
> > > > >No, IQM is a more robust statistic than median, and was developed precisely for situations with few seeds.
> > > >
> > > > Thank you for clarifying this. We have read more in-depth about IQM and now see how it is different from just taking the median since for three seeds we need to calculate a weighted average that includes all seeds. We are currently putting together a new table using IQM.

---

> > ### Author Response · Authors · 2022-11-05
> > **Multiple rollouts in figure 6**
> >
> > > In Figure 6, it would be good to overlay multiple trajectories (instead of just one) so we can see the consistency of the method.
> >
> > Thank you for the suggestion! We have updated figure 6 to include multiple rollouts for each task.

---

> ### Author Response · Authors · 2022-10-27
> **Response**
>
> >Another recurring issue is that a number of design decisions for the proposed algorithm (number of queries, number of ensemble  models, etc.) are specified but not well-justified. It would be good to provide sensitivity analyses for these, as it will help future researchers/users be better equipped for dealing with these types of problems methods.
>
> We added a sensitivity analysis on the number of initial queries and number of queries for ensemble disagreement in the halfcheetah-random-v2 environment in section B of the appendix.
>
> The results are reported with 3 seeds and aggregated using the interquantile mean. We find that more initial queries improve performance for random queries. However, ensemble disagreement performs best with 50 initial queries. Ensemble disagreement outperforms random queries for all numbers of initial queries. The improvement from using ensemble disagreement vs. random queries is less significant for 25 and 100 queries. This is likely due to the fact that 25 queries are not sufficient to train an initial ensemble to estimate the disagreement and  100 queries by itself is sufficient to train a good reward model without additional active learning. 50 is likely strikes a good balance where we have enough to train a good initial ensemble of models while not diminishing the effect of using active queries.
>
> We also vary the number of queries per round and found that ensemble disagreement outperforms random in all settings. The performance for ensemble disagreement is also fairly consistent across the different numbers of queries per round. We note that the more number of queries per round does not necessarily improve performance. However, this is likely due to the fact that other hyper-parameters like the number of iterations are tuned for 10 queries per round.
>
> We are also finalizing a sensitivity analysis on the number of ensemble models and the number of samples for dropout that we believe will address this concern and also provide more insights for future research.

---

> > ### Comment · Reviewer_yMyt · 2022-11-07
> > **Sensitivity analysis**
> >
> > Thanks for adding these. I would suggest adding the discussion you placed in the rebuttal in the paper, as right now it's just the tables so it's harder to make sense of them.
> >
> > Also, it appears Appendix C is empty.

---

> > > ### Author Response · Authors · 2022-11-08
> > > **Sensitivity analysis discussion and formatting**
> > >
> > > Thanks for pointing this out! We have added the discussion to Appendix B and C of our paper and fixed the formatting issue with Appendix C.

---

> ### Author Response · Authors · 2022-10-27
> **Additional sensitivity analyses on number of ensemble models and dropout samples**
>
> > Another recurring issue is that a number of design decisions for the proposed algorithm (number of queries, number of ensemble models, etc.) are specified but not well-justified. It would be good to provide sensitivity analyses for these, as it will help future researchers/users be better equipped for dealing with these types of problems methods.
>
> We have added additional sensitivity analyses on the number of ensemble models and the number of dropout samples in section C of the Appendix.
>
> The default uses 7 ensemble models for both ensemble disagreement and ensemble information gain. For ensemble disagreement, we notice an improvement in performance from increasing the number of ensemble models from 3 to 7 and no improvement and a slight dip in performance from 7 to 14.
> For ensemble information gain, we notice that the performance improves slightly with more ensemble models. However, ensemble information gain seems to be less sensitive to the number of ensemble models compared to ensemble disagreement. This can also mean that ensemble information gain can be effective with fewer ensemble models. Although 14 ensemble models perform slightly better than the default in the case of ensemble disagreement, it does take significantly longer to train due to the additional number of ensemble models to train.
>
> The default number of dropout samples is 30. For dropout disagreement, we notice that with more dropout samples, the performance improves. Dropout info gain on the other hand works best with 30 dropout samples and worse with 15 dropout samples. In both variants, increasing the number of dropout samples beyond 30 does not improve the performance significantly and even slightly degrades performance in the case of dropout information gain.

---

> ### Author Response · Authors · 2022-11-03
> **Updated with new table using IQM**
>
> > In Table 2, since all results are normalized, you should report more robust statistics as suggested in 2 (namely, Interquantile Mean)
>
> Thank you for the suggestion! We added another table in Appendix Section A that gives normalized results using interquartile mean.

---

> > ### Comment · Reviewer_yMyt · 2022-11-07
> > **IQM**
> >
> > Thanks for adding these! It'd be good if you could define IQM before referring to it, as well as cite the IQM paper.

---

> > > ### Author Response · Authors · 2022-11-08
> > > **Updated with IQM Definition**
> > >
> > > Thanks for the response! We have now added an introduction to IQM and a citation to the Agarwal et a. paper in Appendix section A.

---

> > > > ### Public Comment · ~Rishabh_Agarwal2 · 2023-01-07
> > > > **Regarding use of IQM**
> > > >
> > > > Hi authors,
> > > >
> > > > I am curious how the formula used for IQM with 3 seeds is derived. IQM has less uncertainty than median and more robust than mean when used as an aggregate metric, ideally this would correspond to aggregation across both tasks and seeds  -- assuming 16 tasks and 3 seeds, this would correspond to computing aggregate statistics using all the 48 seeds. Ideally, you should also be reporting the statistical uncertainty in these aggregate metrics (i.e. how much variation there is if someone reruns these experiments with 3 seeds).
> > > >
> > > > The library at [https://github.com/google-research/rliable](https://github.com/google-research/rliable) can be used as-is: Just plug-in your numbers and you will get the aggregate plots. See [bit.ly/statistical_precipice_colab](bit.ly/statistical_precipice_colab) for an example colab.
> > > >
> > > > Cheers,
> > > > Rishabh

---

> > > > > ### Author Response · Authors · 2023-02-24
> > > > > **Re IQM**
> > > > >
> > > > > The IQM for three seeds (with returns X <= Y <= Z) is computed using the following formula  (0.25*X + Y + 0.25*Z) / (1.5). Thanks for the link to the code repo. It looks really useful for uncertainty analysis!

---

### Author Response · Authors · 2022-11-11
**Summary of Changes**

We’d like to thank the reviewers for the suggested changes.
We have included a summary of the changes we have made to the paper so far
- Updated Figures 5 and 6 to have a white background to improve the visibility of trajectories
- Overlaid 4 trajectories instead of just one in Figure 6 to show the consistency of our method
- Included a discussion on Castro, Li, & Zhang, 2019; Inverse Reinforcement Learning with Multiple Ranked Experts. Paper in our related work section
- Sensitivity analyses on the number of initial queries and number of queries for ensemble disagreement in the halfcheetah-random-v2 environment in section B of the appendix as well as a discussion.
- Sensitivity analyses on the number of ensemble models and the number of dropout samples in section C of the Appendix as well as a discussion.
- Added another table in Appendix Section A that gives normalized results using interquartile mean on the performance of OPAL using reward predicted after N rounds of querying.
- Added a paragraph in 4.2 called “Searching the Offline Dataset for Informative Queries” where we discuss efficiency.
- Add justification, for why estimating/representing uncertainty is important in our work in Section 4
- Add a clarification that our analysis of offline RL benchmarks shows that many are ill-suited for evaluating any kind of reward learning in the abstract.
- Add a clarification that our new benchmarks are specifically designed for offline preference-based reward learning.
- Add clarification in the last paragraph of page 2 why we seek tasks where the choice of reward function has a strong effect on performance. Since otherwise we cannot tell whether one method for learning rewards is better or worse than another.


We appreciate the reviewers' suggestions and believe that the paper has been significantly improved. Please let us know if there is anything else that should be changed for the paper to be accepted.

---

### Decision · Action_Editors · 2022-11-28

**Recommendation:** Accept with minor revision

**Comment:**

The paper is well-written and clear. The results are convincing and the experiments are well-conducted. The subject and the topic this paper is studying is interesting and important. The authors were happy about the paper and leaning accept due to some revisions that they requested. My comments are based on an earlier version of the paper, but it seems like the authors have already addressed some of the comments raised by the reviewers. As a result, I recommend this paper for acceptance with minor revisions. However, there are some small minor revisions that needs to be done as pointed out by the reviewers (if they are not already addressed). Some important ones are:
1. Sensitivity analysis on the hyper-parameters chosen to justify as noted by the reviewer yMyt.
2. The phrase *"We are the first to evaluate Dropout Disagreement..."* is unnecessary. Both reviewers yMyt and 2gDE  suggested to remove it.
3. Efficiency analysis of identifying the best pair of trajectories to compare as pointed out by reviewer 2gDE.
4. Comparisons to T-REX and other reward learning baselines as suggested by reviewer Ndnj.
5. Some small typos, and cosmetic changes as pointed out by the reviewers.

**Audience:**

The paper looks into the problem of learning reward models from human preferences from offline datasets. This is a very important problem that needs to be addressed to be able to deploy offline RL algorithms into real-world.  As a result of growing prominence of offline RL and reward learning within the community. The paper also outlines several interesting results in offline policy optimization and reward learning. As a result, this paper would be interest of the TMLR audience.

**Claims And Evidence:**

This paper proposes an approach called *Offline Preference-based Apprenticeship Learning* (OPAL) that can make use of offline data for reward learning with human preferences.  Given an offline dataset, OPAL:

1. Firstly queries the human raters for preference labels over trajectory segments from the dataset.
2. Learn a reward model on the human preferences.
3. Learn a policy with offline RL over the dataset relabelled with the learned reward model.
The paper also proposes a method to compute uncertainty for the reward models to actively select informative queries and the paper found out that ensemble-based disagreement queries outperform other baselines.

The proposed approach and the methodology followed in the paper are sound and valid. The experiments are conducted across a wide range of environments and datasets. The reviewers and I couldn't find any major flaws in the results or experimentations. The conclusions and the data interpretation of the experiments are valid and reliable.